**EMBO** *reports*

# Single-cell analysis of signalling and transcriptional responses to type I interferons

Rachel E Rigby [1], Kevin Rue-Albrecht [2], Aleksandr Fedorov [1], David Sims[2] & Jan Rehwinkel [1]✉

## Abstract

Type I interferons (IFNs) play crucial roles in antiviral defence, autoinflammation and cancer immunity. The human genome encodes 17 different type I IFNs that all signal through the same receptor. Non-redundant functions have been reported for some type I IFNs. However, whether different type I IFNs induce different responses remains largely unknown. Here, we stimulate human peripheral blood mononuclear cells (PBMCs) with recombinant type I IFNs to address this question in multiple types of primary cells. We analyse signalling responses by mass cytometry and changes in gene expression by bulk and single-cell RNA sequencing. We find cell-type specific changes in the phosphorylation of STAT transcription factors and in the gene sets induced and repressed upon type I IFN exposure. We further report that the magnitude of these responses varies between different type I IFNs, while qualitatively different responses to type I IFN subtypes are not apparent. Taken together, we provide a rich resource mapping signalling responses and IFN-regulated genes in immune cells.

**Keywords** Type I Interferon; PBMCs; Interferon Stimulated Genes; Mass Cytometry; Single-cell RNAseq
**Subject Categories** Immunology; Methods & Resources; Signal Transduction

## Introduction

Type I IFNs are a family of cytokines essential to antiviral immunity (McNab et al, 2015). Typically, type I IFNs are produced transiently following an infection. However, in some autoimmune and autoinflammatory conditions, type I IFNs are secreted chronically and initiate and/or exacerbate disease (Crow and Manel, 2015). Furthermore, type I IFNs impact on bacterial infections and cancer, with both beneficial and detrimental effects (McNab et al, 2015; Zitvogel et al, 2015). Therefore, type I IFNs have wide-ranging effects in the immune system and in diseases.

Type I IFNs were identified in the 1950s for their ability to "interfere" with virus infection (Isaacs and Lindenmann, 1957; Nagano and Kojima, 1954). Much research has been done since on

type I IFNs and led to the concept of the type I IFN system (Fig. 1A) (Stark and Darnell, 2012). This host defence response is triggered when cells autonomously detect virus infection. Nucleic acids are often a molecular signature of viral infection. Nucleic acid sensors recognise viral DNA or RNA, or disturbances to the homeostasis of cellular nucleic acids, such as mis-localisation (Hartmann, 2017). These sensors then initiate signalling cascades resulting in the expression and secretion of type I IFNs. Like in other species, there are many different type I IFNs in humans: 13 IFN-α sub-types, IFN-β, -ω, -ε and -κ. Type I IFNs engage IFNAR, a receptor expressed on the surface of all nucleated cells (Platanias, 2005). IFNAR is composed of two subunits, IFNAR1 and IFNAR2, which are associated at their cytoplasmic tails with the kinases TYK2 and JAK1, respectively. The canonical signalling pathway downstream of IFNAR involves the formation of the ISGF3 complex by phosphorylated STAT1 and STAT2 together with IRF9. ISGF3 binds to IFN-stimulated response elements (ISREs) in the promoters of type I IFN-responsive genes and thereby drives the expression of IFN-stimulated genes (ISGs). Non-canonical signalling pathways downstream of IFNAR can involve the phosphorylation of other members of the STAT transcription factor family (STAT3-6), members of the MAP kinase family, components of the PI3K/mTOR pathway, or NF-κB (Fig. 1A) (de Weerd and Nguyen, 2012; He et al, 2020; Mazewski et al, 2020; Platanias, 2005; Schreiber, 2017; Schreiber and Piehler, 2015; Yang et al, 1996). Canonical and non-canonical IFNAR signalling profoundly affect the transcriptome (Rusinova et al, 2013). Initial conservative estimates suggested that ~400 genes are induced by type I IFN (Schoggins et al, 2011) and more recent experiments in human fibroblasts found that ~10% of transcripts are controlled by type I IFN (Shaw et al, 2017). Moreover, type I IFNs also downregulate mRNA levels of some genes (Guo et al, 2020; Shaw et al, 2017). Some ISGs encode virus restriction factors, cellular proteins that block infection, while others encode proteins involved in virus detection or in cellular and adaptive immune responses (Malim and Bieniasz, 2012; Schoggins, 2019). Collectively, the genes regulated by type I IFNs thus complete the "loop" of the type I IFN system (Fig. 1A).

All type I IFNs bind to and signal through IFNAR, raising the question whether these cytokines have redundant functions. Multiple explanations likely exist for the large number of type I IFNs. These include tissue-specific functions. Indeed, IFN-ε is expressed in the female reproductive tract and may play a role in cancer immunosurveillance (Barriga et al, 2022; Fung et al, 2013).

[1]MRC Translational Immune Discovery Unit, MRC Weatherall Institute of Molecular Medicine, Radcliffe Department of Medicine, University of Oxford, Oxford, UK. [2]MRC WIMM Centre for Computational Biology, MRC Weatherall Institute of Molecular Medicine, University of Oxford, Oxford, UK. ✉E-mail: jan.rehwinkel@imm.ox.ac.uk

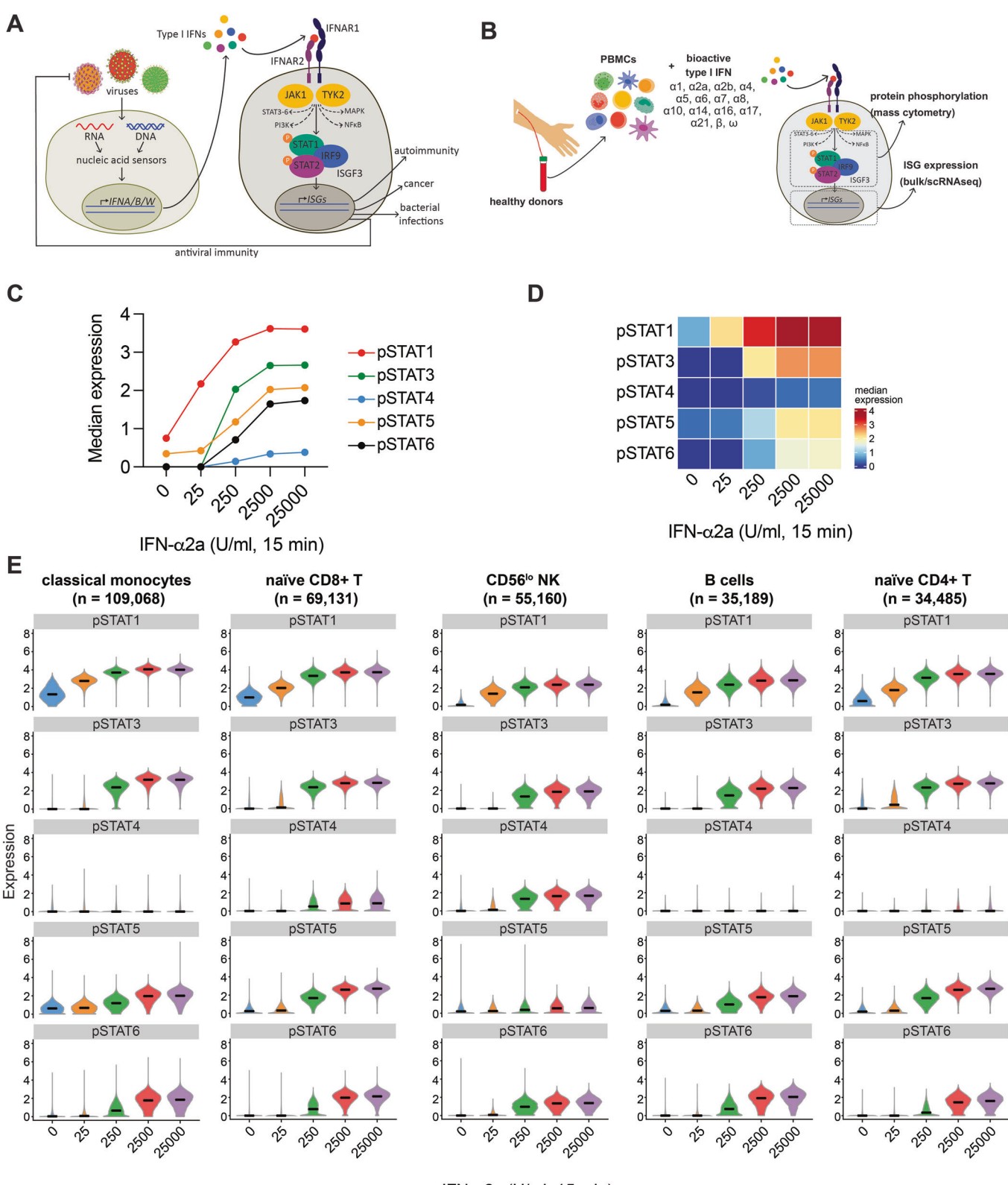

IFN-κ is expressed in keratinocytes and controls repair of skin wounds (LaFleur et al, 2001; Wolf et al, 2022). However, all other type I IFNs are expressed upon stimulation by most nucleated cells (Pestka et al, 2004). Early work suggested that the ability of IFN-α

sub-types to enhance NK cell cytotoxicity varies over several orders of magnitude (Ortaldo et al, 1984). More recently, several studies reported different potencies of type I IFNs in blocking viral infections and in exerting anti-tumour effects (Buzzai et al, 2020;

◄ **Figure 1.  Phosphorylation of STAT proteins in response to IFN-α2a.**

(A) Overview of the type I IFN system. (B) Schematic representation of the mass cytometry workflow to quantify protein phosphorylation in response to type I IFN-treatment. (C) Median expression of pSTAT1, pSTAT3, pSTAT4, pSTAT5 and pSTAT6 in PBMCs treated with increasing concentrations of IFN-α2a for 15 min. (D) Depiction of the data shown in (C) as a heatmap. (E) Violin plots showing expression of each pSTAT in the indicated cell types in response to increasing concentrations of IFN-α2a. The number of cells analysed is shown in parentheses. Violin plots for all cell types are shown in Appendix Fig. S1C. Data are from one experiment with one donor. See also Appendix Figs. S1 and S2 and Datasets EV1 and EV2.

Harper et al, 2015; Karakoese et al, 2022; Lavender et al, 2016; Matos et al, 2019; Rout et al, 2022; Schmitz et al, 2022; Schuhenn et al, 2022; Tauzin et al, 2021; Xie et al, 2022). Transcriptomic analysis indicates that different type I IFNs induce common and unique genes in CD4 T cells (Guo et al, 2020). These differences may be due to the observation that type I IFNs have varying characteristics of receptor binding (de Weerd et al, 2013; Thomas et al, 2011). Combined with cell type-specific expression levels and stoichiometries of proteins involved in IFNAR signalling, different cell types have been suggested to respond in distinct ways to different type I IFNs. Indeed, mouse IFN-αA induces common and unique ISGs in different murine immune cells (Mostafavi et al, 2016). Moreover, human B and CD4 T cells respond more strongly to IFN-α2 than to IFN-ω, while monocytes show equivalent responses (Thomas et al, 2011). However, systematic studies comparing the effects of different type I IFNs in multiple types of primary cells are lacking at present.

Here, we close this knowledge gap using single-cell technologies. We observed cell-type-specific signalling and transcriptomic responses to type I IFN. Surprisingly, however, our comparison of multiple type I IFNs revealed only quantitative, but no apparent qualitative, differences. Our data mapping signalling and transcriptomic responses to type I IFNs represent a resource valuable to multiple scientific fields and are easily accessible through an interactive online interface (https://rehwinkellab.shinyapps.io/ifnresource/).

## Results

### Mass cytometry analysis reveals cell-type-specific STAT response patterns to type I IFNs in PBMCs

Systematic studies of the signalling pathways activated upon engagement of IFNAR with different type I IFNs in different cell types are currently lacking. To investigate this in multiple primary human cell types, we established a mass cytometry workflow to quantify cellular activation by phospho-protein staining in freshly isolated human peripheral blood mononuclear cells (PBMCs) stimulated with recombinant type I IFN (Fig. 1B). We chose this model because of (i) the possibility of treating primary cells with minimal disturbance, (ii) the plethora of cell types found among PBMCs and ease of access to blood, (iii) the availability of bioactive human recombinant type I IFNs and (iv) the potential use of results as future biomarkers or for treatment strategies. We developed a 39 antibody staining panel comprising of 26 lineage markers to allow identification of approximately 20 major cell subsets and 13 antibodies to interrogate IFNAR signalling. The latter included antibodies against IFNAR1 and IFNAR2, nine phospho-proteins (pSTAT1, pSTAT3, pSTAT4, pSTAT5, pSTAT6, pERK1/2, pp38,

pMAPKAPK2, pNFκB) and total STAT1 and STAT3. A suitable antibody for pSTAT2 was not available at the time this study began.

Phosphorylation of STAT proteins occurs rapidly and peaks around 30 min after IFNAR engagement (Schreiber and Piehler, 2015). Here, we stimulated cells for 15 min to avoid saturation effects. To establish a suitable dose range, we stimulated PBMCs with 0, 25, 250, 2500 or 25,000 U/ml of IFN-α2a and assessed the phosphorylation of STAT proteins. IFN-α2a was chosen because it is the IFN-α subtype most commonly used both in vitro and in vivo. For dosing, we used bioactivity (U/ml), as determined by the manufacturer of the recombinant protein by inhibition of the cytopathic effect of encephalomyocarditis virus in A549 cells, instead of mass (pg/ml), because recombinant type I IFN preparations may contain inactive protein. All STAT proteins were phosphorylated in response to IFN-α2a stimulation in a dose-dependent manner, saturating above the 2500 U/ml dose (Fig. 1C,D). STAT1 phosphorylation was the most sensitive response to stimulation, increasing even at the lowest dose used. In contrast, increased phosphorylation of the other STATs required stimulation with at least 250 U/ml IFN-α2a.

We next quantified the response to IFN-α2a in different cell types, taking advantage of our detailed phenotyping panel. Cells were clustered using the phenotyping markers and the resulting clusters were manually annotated, giving 17 distinct populations (Appendix Fig. S1A,B). The expression of each pSTAT protein was then plotted for each cell population (Appendix Fig. S1C, Datasets EV1 and EV2). The most abundant cell types (classical monocytes, naive CD8 + T cells, CD56$^{lo}$ NK cells, B cells and naive CD4 + T cells) are shown in Fig. 1E. Baseline levels of pSTAT1 varied between different cell types, with myeloid cells and central memory T cells having higher expression. Levels of pSTAT5 were highest in myeloid cells and plasmablasts at baseline whereas pSTAT3, 4 and 6 were undetectable in unstimulated cells (Appendix Figs. S1 and S2). STAT1 was phosphorylated in all cell types, in line with its role in canonical signalling downstream of IFNAR. STAT3, STAT5 and STAT6 phosphorylation was also detected in all cell types, although only at concentrations of 250 U/ml IFN-α2a and above. In contrast, STAT4 phosphorylation only occurred in some T cell subsets and NK cells, reflecting the largely T cell and NK cell restricted expression of STAT4 (Uhlen et al, 2019). This provides an explanation for the lower pSTAT4 response we observed when we analysed all cells together (Fig. 1C). Notably, there was considerable heterogeneity in the extent of STAT phosphorylation within each cell population (Fig. 1E; Appendix Fig. S1C).

These data show that our mass cytometry workflow successfully and simultaneously quantified phosphorylation of multiple proteins in response to stimulation with type I IFN. Our results further highlight that different types of cells respond in different ways to type I IFN, and that cell-type-specific responses may be masked by bulk analysis.

## Different type I IFNs induce qualitatively similar transcription factor phosphorylations

We next used our mass cytometry workflow to investigate the response to different type I IFNs. Cells were stimulated for 15 min with 2500 U/ml of commercially available bioactive recombinant protein for the 13 IFN-α subtypes, IFN-β and IFN-ω. IFN-ε and IFN-κ were not included as their expression is restricted to specific tissues (female reproductive tract and skin, respectively) (Fung et al, 2013; LaFleur et al, 2001). The concentration was chosen to ensure that phosphorylation events of all STAT proteins could be detected. All type I IFNs induced phosphorylation of the STAT proteins when all cells were analysed together (Fig. 2A,B). Analysis of individual cell populations revealed that, for any given cell type, all type I IFN subtypes led to phosphorylation of STAT1 to a similar extent, although clear differences in pSTAT1 levels were evident between different cell types (Fig. 2C,D, Datasets EV3 and EV4). For pSTAT3 (Fig. 2E), pSTAT4 (Fig. 2F), pSTAT5 (Fig. 2G) and pSTAT6 (Fig. 2H), quantitative differences were evident between type I IFN subtypes. For example, IFN-α1 induced lower levels of pSTAT4 and IFN-α1, -α2b and -α14 induced lower levels of pSTAT6. However, qualitative differences in the response to different type I IFN subtypes were not apparent amongst the cell types analysed. Similar results were obtained for phosphorylation of NFκBp65, p38, MAPKAPK2 and ERK1/2 (Appendix Fig. S3).

To investigate the effects of different type I IFNs on phosphorylation of STAT proteins at later time points, we first stimulated PBMCs for 90 min with increasing doses of a subset of five type I IFNs (IFN-α1, -α2a and -α10; IFN-β; IFN-ω). These represent low, medium and high potency IFN-α subtypes (Fig. 2), together with the unique IFN-β and IFN-ω. At this time point, phosphorylation levels of all STATs were lower compared to the 15 min timepoint, with only small increases even at 2500 U/ml (Appendix Fig. S4). Quantitative differences in the amount of each type I IFN required to increase STAT phosphorylation were evident, with higher doses of the three IFN-α subtypes required compared to IFN-β and IFN-ω (Appendix Fig. S5). Increased baseline levels of pSTAT1 and pSTAT3 were seen at this timepoint (Appendix Fig. S4), likely a consequence of the increased time cells spent in culture before analysis. Stimulation with type I IFNs increased phosphorylation of STAT1 in all cell types; however, as observed 15 min after stimulation, differences between type I IFN subtypes were largely quantitative (Appendix Fig. S5D, Datasets EV5 and EV6). Only slight increases in phosphorylation of the other STAT proteins were evident at this time point and again differences between type I IFN subtypes were quantitative (Appendix Fig. S5E–H). Similar results were obtained when PBMCs were stimulated for 24 h (Appendix Fig. S6, Datasets EV7 and EV8).

Taken together, these data show that in PBMCs stimulated ex vivo, different recombinant type I IFNs induce similar profiles of transcription factor phosphorylation, although the magnitude of the response varies between type I IFNs.

## Different type I IFNs induce qualitatively similar expression changes of coding and non-coding RNAs in PBMCs

IFNAR signalling profoundly alters the transcriptome. In addition to the induction of ISGs, type I IFNs may downregulate mRNA levels of some genes (Guo et al, 2020; Shaw et al, 2017). We stimulated PBMCs from three donors with 250 U/ml type I IFN, using the set of five type I IFNs described above. 24 h after stimulation, we extracted RNA and, following enrichment of polyadenylated transcripts, performed RNA sequencing. The 24 h timepoint was chosen based on RT-qPCR analysis of the induction of seven ISGs over time in response to IFN-α1 and IFN-β. Although the levels of some ISGs peaked 2–4 h after stimulation, these transcripts were still upregulated after 24 h, a timepoint at which other ISGs were maximally induced (Appendix Fig. S7). Therefore, the 24 h timepoint likely allows detection of early and late responding ISGs.

Sequence reads were aligned to the genome and used to select reliably supported RNA models from the GENCODE annotation. We then quantified the abundance of these RNAs and grouped technically indistinguishable transcripts to ensure the robustness of the expression estimate. These transcript groups are hereafter referred to as 'transcripts' for brevity. We performed our analysis at transcript level to account for genes such as NCOA7 or ADAR1 that encode both type I IFN inducible and IFN-nonresponsive transcripts. Based on these data, we confirmed that the expression of key IFNAR signalling components was comparable across all three donors (Appendix Fig. S8A; Datasets EV9 and EV10). Of note, the expression of STAT5A and STAT5B individually could not be reliably estimated due to high sequence similarity. Next, we statistically compared the expression of transcripts mapping to coding genes in cells stimulated with each type I IFN against unstimulated samples. Here, each IFN induced widespread differential expression, with more transcripts being upregulated than downregulated (Fig. 3A; Dataset EV11). IFN-β induced the largest number of differentially expressed transcripts (DETs, Appendix Fig. S8B), while IFN-α1 induced the fewest. Overall, upregulated transcripts exhibited the greatest fold changes and statistical significance (Fig. 3A; Appendix Fig. S9). Applying a 1.5-fold change threshold to capture moderate changes in transcript levels and a significance level of adj. $p$-value $\leq 0.05$, we identified 4585 RNAs that were upregulated and 3343 that were down-regulated in response to at least one type I IFN (Fig. 3C). Among these, 1240 were consistently upregulated and 103 were consistently downregulated by all tested type I IFNs (Appendix Fig. S10A).

Gene Ontology (GO) analysis confirmed an expected enrichment for terms such as "Defense Response to Virus" and "Negative Regulation of Viral Process" among genes associated with universally up-regulated transcripts (Appendix Fig. S11A). Using a published list of 379 human ISGs compiled from microarray data from multiple cell lines and tissues (Schoggins et al, 2011) (Dataset EV12), we found that 345 had at least one isoform robustly detected in our RNAseq dataset, and 298 hosted RNAs up-regulated by at least one IFN in PBMCs (Fig. 3D). PBMCs from all three donors showed comparable induction of most RNAs, with transcripts mapping to canonical ISGs being among the most up-regulated (Fig. 3E).

In addition to inducing the expression of many RNAs, type I IFNs also significantly repressed the expression of another set of transcripts (Fig. 3C). We used GO analysis to gain insight into biological processes related to the coding genes corresponding to RNAs significantly down-regulated by all type I IFNs (Appendix Fig. S11B). This analysis showed only weakly enriched GO categories, each containing only a few down-regulated genes.

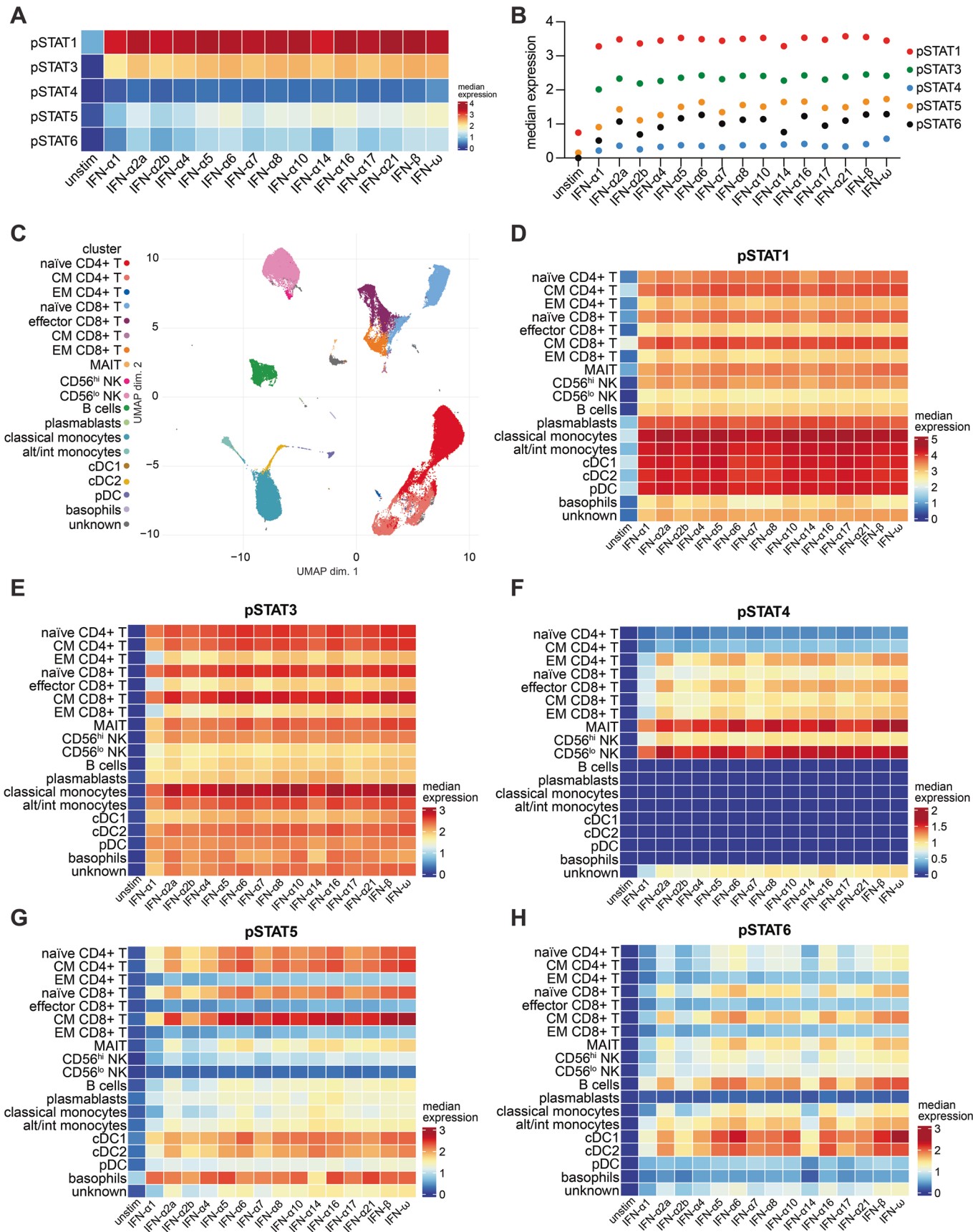

◄ **Figure 2. Phosphorylation of STAT proteins in response to different type I IFNs.**

(A) Median expression of pSTATs in PBMCs in response to treatment with 2500 U/ml of the indicated type I IFNs for 15 min. (B) Depiction of the data shown in (A) as a dot plot. (C) UMAP plot showing clustering of PBMCs and identification of different cell types based on expression of the phenotyping markers. (D–H) Heatmaps showing median expression of pSTATs in each cell type in response to treatment with different type I IFNs. Data are from one experiment with one donor. See also Appendix Figs. S3–S6 and Datasets EV3–EV8.

We also considered transcripts that met our criteria for significance (adj. $p$-value ≤ 0.05, 1.5 fold up- or down-regulated) for only one type I IFN. The bulk of these were found in response to IFN-β (Appendix Fig. S10). Visualisation of the expression of these RNAs across all samples using heatmaps suggested that, for the vast majority of these transcripts, the differences in expression between the different type I IFNs were quantitative (Appendix Fig. S10). For example, RNAs that were only up-regulated above threshold for IFN-β stimulation showed a clear trend towards induction by the other four type I IFNs tested, albeit to lower levels (Appendix Fig. S10).

In addition to reads mapping to protein-coding genes, we also examined those mapping to long non-coding RNAs (lncRNAs) and pseudogenes, here referred to as non-coding RNAs (ncRNAs) for simplicity. The majority of lncRNAs are polyadenylated (Derrien et al, 2012), and some pseudogenes either retain functional polyadenylation sites or possess poly-A tails from retrotransposition of mature RNAs, enabling their detection in our poly-A enriched cDNA libraries. Overall, statistical analysis revealed that, similar to RNAs mapping to coding loci, many non-coding transcripts were repressed or induced in response to type I IFNs, with more ncRNAs being up-regulated than down-regulated and IFN-β having the strongest effect (Fig. 3B; Appendix Figs. S8C and S12, Dataset EV13). A total of 85 noncoding transcripts were universally induced and 11 repressed by all type I IFNs (adj. $p$-value ≤ 0.05, fold change of at least 1.5) (Appendix Fig. S12C).

Most universally regulated ncRNAs are uncharacterised, though exceptions include transcripts from the microRNA host gene (HG) *MIR4432HG* and multiple antisense (AS) or divergent (DT) transcripts. Relaxing the criteria to regulation by at least one tested IFN revealed four up-regulated and four down-regulated microRNA precursors (Appendix Fig. S12D), including MIR155HG, a known positive regulator of the innate antiviral response (Rai et al, 2022). Using the same criteria, we found that roughly half of divergent ncRNAs (19 of 39) were associated with transcripts from protein-coding genes that were themselves differentially expressed in response to at least one IFN, with most of these RNAs co-regulated with their matched coding neighbours (Appendix Fig. S12E). Similarly, almost half (54 of 132) of antisense DETs had matched protein-coding DETs, and among these, 18 were anticorrelated with their sense partners, suggesting AS-RNA-mediated post-transcriptional repression (Appendix Fig. S12F).

To further understand the regulatory basis underlying the observed transcriptional changes, we examined promoters associated with coding and non-coding DETs. We matched DETs to promoter-like sequences from the ENCODE registry (Consortium et al, 2020), imputing promoter locations when direct matches were unavailable. De novo motif analysis using STREME (Bailey, 2021) revealed that ISRE-like motifs were the only motifs significantly enriched in the promoters of IFN-stimulated RNAs, compared to robustly expressed but unregulated transcripts (Appendix Fig. S13A). We found no motifs that reliably distinguished transcripts regulated by individual type I IFN subtypes or that differentiated downregulated transcripts from baseline RNAs. To further explore transcriptional regulation, we investigated transcription factors (TFs) with known binding motifs catalogued in the JASPAR database (Rauluseviciute et al, 2024). Here, we scored all promoters against clustered JASPAR motifs and Z-standardised the final scores relative to all GENCODE high-confidence transcripts, marking RNAs expressed in PBMCs as potentially regulated (Z-score ≥ 2) or likely unregulated (Z-score ≤ −1). We then compared transcriptional responses between these two groups, hypothesising that actively involved TFs would show significant associations between motif scores and expression changes. Consistent with the STREME results, only ISRE-like TF motifs were strongly associated with IFN-induced expression changes across all IFN subtypes, with ISRE-positive transcripts exhibiting the greatest expression changes in IFN-β-treated samples (Appendix Fig. S13B). Other significant, though less impactful, associations included GAS-like motifs, recognised by STAT1 dimers in response to IFN-γ stimulation; IRF6; ZNF93, previously shown to repress L1 elements (Jacobs et al, 2014); ZFP57; and NRF1, a transcription factor shown to act as a negative regulator of innate antiviral signalling (Zhao et al, 2023). These TFs were consistently associated with expression changes across all tested IFNs and were expressed in PBMCs (Appendix Figs. S8A and S13C), with the exception of ZFP57, which was undetectable in one donor.

Taken together, these data reveal a robust and broadly similar transcriptional response in primary human PBMCs following 24 h of stimulation with five different type I IFNs. Although IFN-β consistently elicited the strongest response, the differences observed among the IFN subtypes were primarily quantitative rather than qualitative. This pattern extended to the non-coding transcriptome, including antisense RNAs and microRNA precursors, suggesting the activation of a shared regulatory programme. We also found that among all JASPAR-annotated transcription factors, only ISRE-like motifs were strongly associated with observed expression changes, further indicating that all tested type I IFNs converge on a single transcriptional pathway. Therefore, the type I IFNs examined here appear to differ principally in the magnitude, not the nature, of their regulatory effects in human PBMCs. It is possible that quantitative differences result from differences in affinities or half-lives of the STAT and/or IRF complexes formed after engagement of IFNAR by different type I IFNs.

## Single cell level analysis identifies 'core' ISGs induced by type I IFN in all types of PBMCs

PBMCs represent a heterogeneous mix of multiple different cell types of varying frequencies. Given that different cell types displayed different signalling responses to type I IFN stimulation

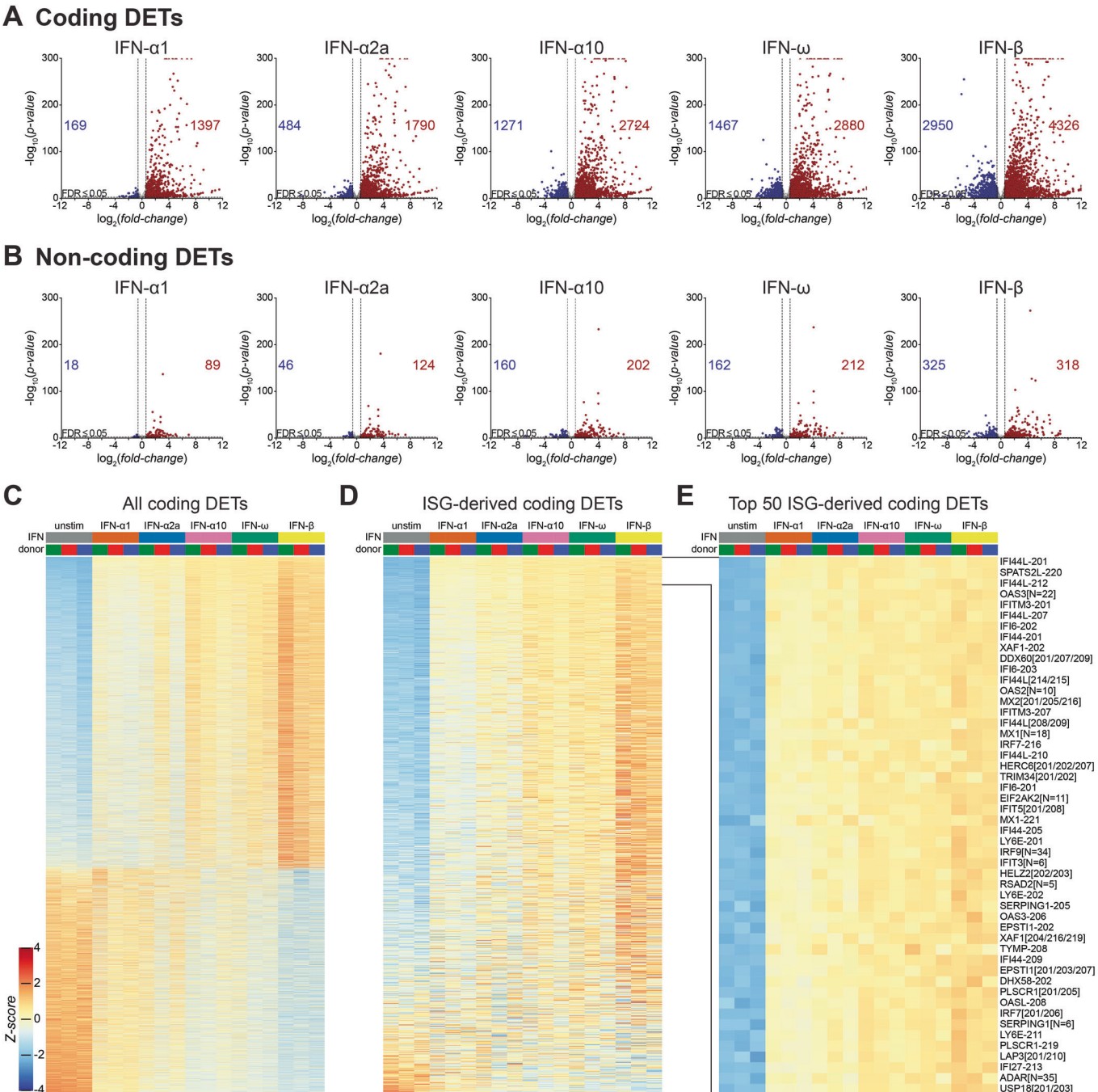

**Figure 3. Bulk RNA-seq of type I IFN-stimulated PBMCs.**

(A, B) Volcano plots of differentially expressed transcripts mapping to protein-coding (A) and non-coding (B) genes in PBMCs stimulated with 250 U/mL of the indicated type I IFNs for 24 h, compared to mock-stimulated PBMCs ($n = 3$ matched biological replicates). Statistical significance was assessed using a two-sided Wald test (DESeq2). Red and blue indicate transcripts significantly up-regulated or down-regulated (Benjamini-Hochberg adjusted $p$-value ≤ 0.05; fold change ≥ 1.5), respectively. The total number of differentially expressed transcripts (DETs) are shown. (C) Heatmap of all transcripts mapping to protein-coding genes that are differentially expressed in response to at least one type I IFN (4585 up- and 3343 down-regulated). (D, E) Heatmaps showing expression of coding DETs mapping to annotated ISGs (D) and the top 50 upregulated ISG-derived transcripts (E). Transcript names follow Ensembl nomenclature. For transcripts belonging to technically indistinguishable groups, shared prefixes are followed by unique suffixes in brackets, separated by slashes (e.g., RNA[201/207]); for groups with more than five indistinguishable RNAs, the total number is indicated (e.g., RNA[N = 35]). In (C–E), transcripts are ranked by Z-score across all type I IFN-stimulated samples. Data are from one experiment with three donors. See also Appendix Figs. S7–S13 and Datasets EV9–EV13.

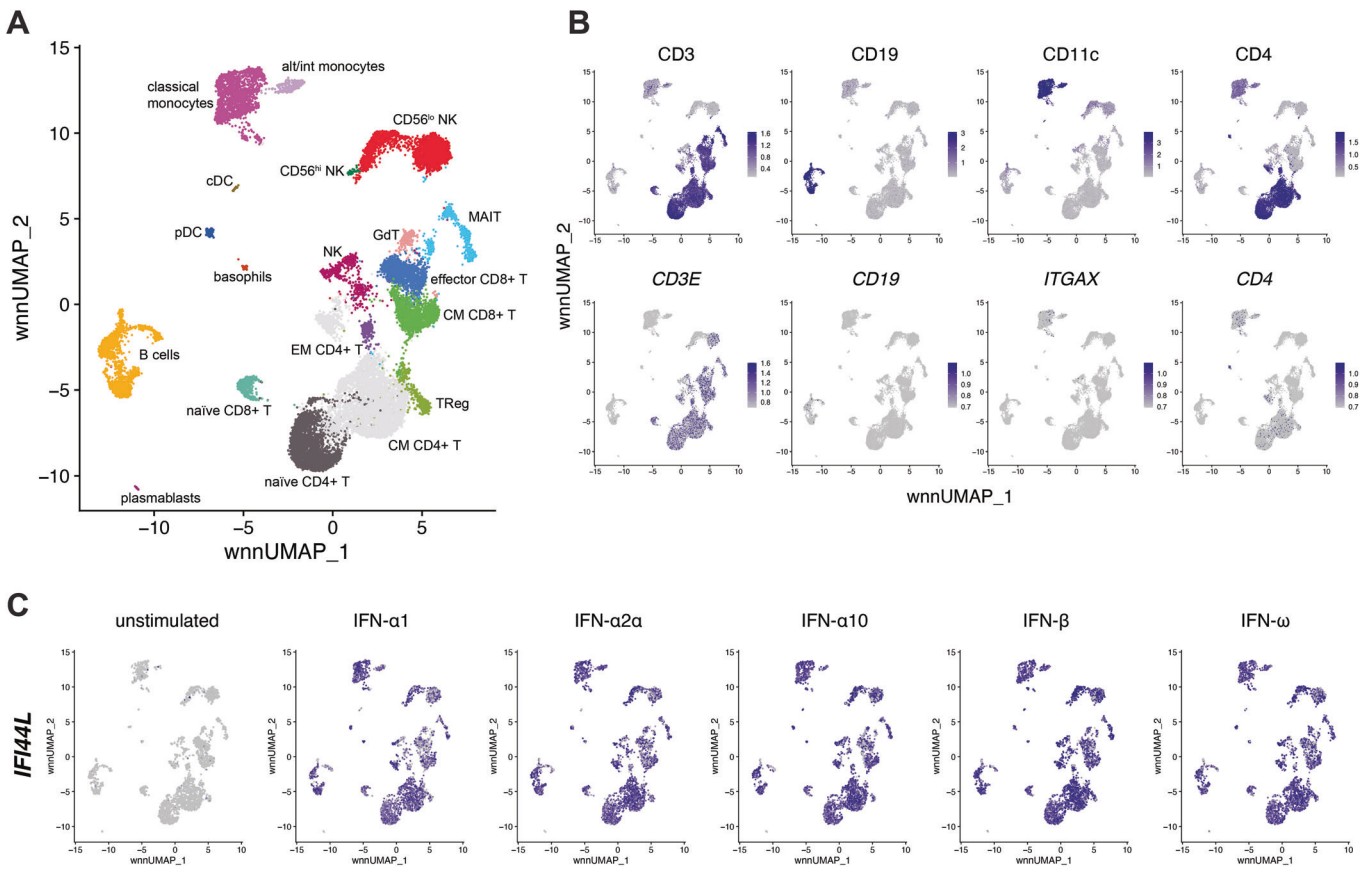

**Figure 4. Single-cell RNAseq analysis of PBMCs stimulated with type I IFNs.**

(A) UMAP visualisation of 19,587 single cells clustered following weighted nearest neighbour (WNN) analysis integrating protein and RNA data for each cell. (B) UMAP plots showing expression of the phenotyping markers CD3, CD19, CD11c and CD4 at the protein (top row) and RNA (bottom row) level. (C) UMAP showing expression of *IFI44L* in unstimulated and type I IFN-stimulated samples. Data are from one experiment with one donor. See also Appendix Figs. S14 and S15 and Datasets EV14–EV18.

(Figs. 1 and 2) and the possibility of cell-to-cell variation, we used scRNAseq to further analyse transcriptional responses to type I IFNs. We treated PBMCs from one donor with 250 U/ml IFN-α1, -α2a, -α10, -β or -ω for 24 h. We took advantage of the utility of our mass cytometry phenotyping panel by staining cells with CITE-seq antibodies corresponding to 20 of these markers prior to capture on a 10X Genomics platform. In brief, CITE-seq involves antibodies conjugated to nucleic acid-based barcodes with a polyA tail, which are sequenced alongside mRNA from captured single cells (Stoeckius et al, 2017). After demultiplexing and quality control, the dataset consisted of 19,587 single cells, with an approximately equal split between samples (unstimulated = 2690 cells, IFN-α1 = 3328, IFN-α2a = 3426, IFN-α10 = 3332, IFN-β = 3350, IFN-ω = 3461). Seurat weighted nearest neighbour (WNN) analysis was used to cluster cells based on mRNA and protein expression (Hao et al, 2021) and cells were visualised using UMAP (wnnUMAP) (Fig. 4A; Appendix Fig. S14). Cell types were identified based on expression of protein markers, in combination with mRNA expression of known markers not present in the CITE-seq antibody panel. A number of markers were detected equally well at protein and mRNA levels, such as CD3, but mRNAs for other markers including CD19, CD11c and CD4 were not detectable (Fig. 4B).

To identify genes induced in response to stimulation with each type I IFN, we performed differential expression analysis for each cell type, comparing type I IFN-treated samples to the unstimulated sample (Datasets EV14–EV18). This was only possible for cell types with an average of >50 cells per sample, precluding this analysis for pDCs, basophils, CD56hi NK cells, cDCs and plasmablasts. We identified only 10 genes that were significantly up-regulated in all cell types in response to all five type I IFN subtypes tested here: *IFI44L, ISG15, IFIT3, XAF1, MX1, IFI6, IFIT1, TRIM22, MX2* and *RSAD2* (Fig. 4C; Appendix Fig. S15). We term these genes that universally mark the type I IFN response 'Core ISGs'.

## Myeloid and lymphoid cells express shared and cell type-specific ISGs

Next, we investigated whether ISGs were regulated in a cell type-specific manner. We first established how many genes were significantly up-regulated by monocytes (classical monocytes, alternative/intermediate monocytes) and lymphocytes (central memory CD4 + T cells, naive CD4 + T cells, CD56lo NK cells, B cells, central memory CD8 + T cells, effector CD8 + T cells, MAIT cells, other NK cells, naive CD8 + T cells, regulatory T cells, effector memory CD4 + T cells and γδ-T cells) in response

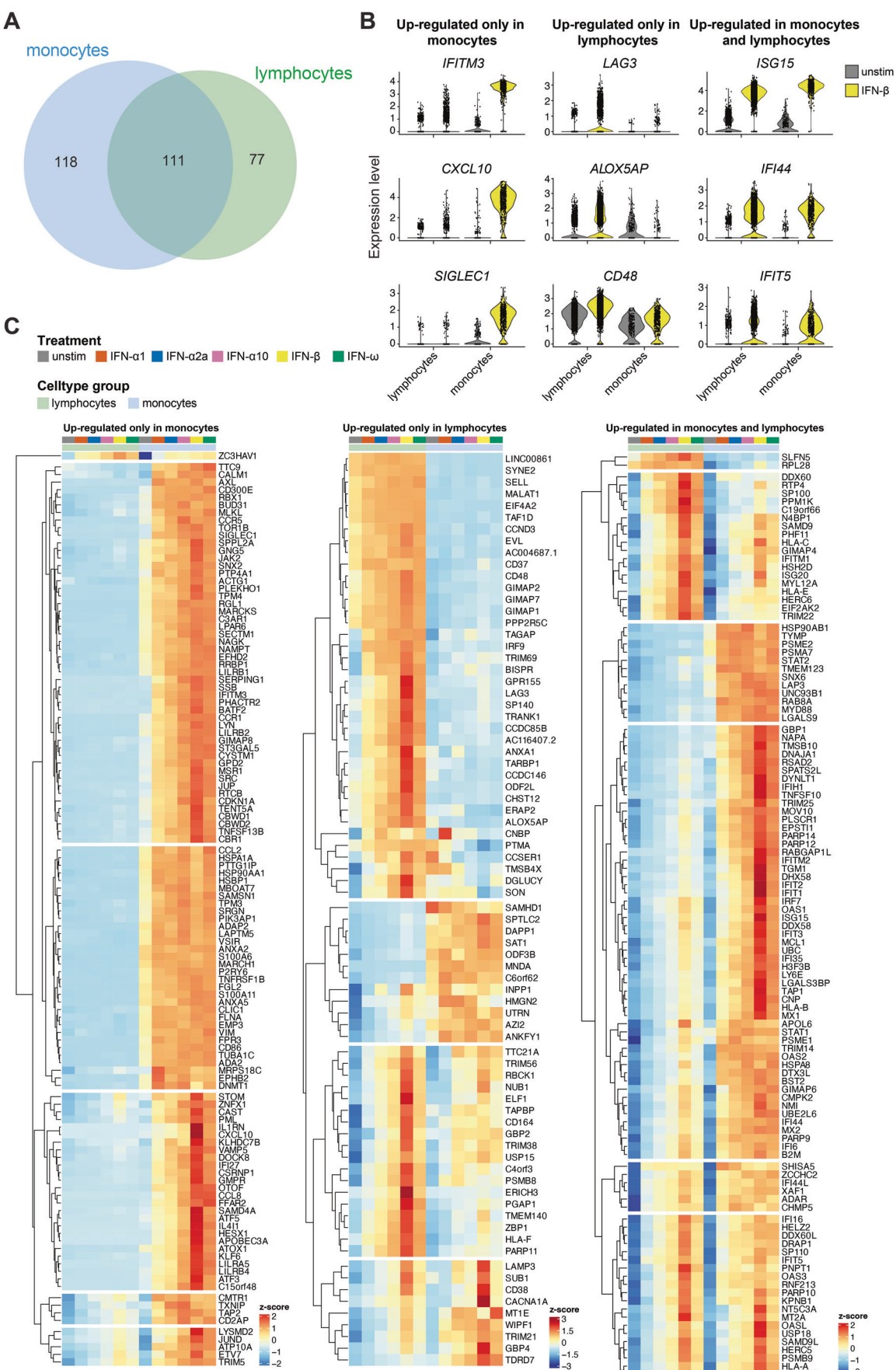

**Figure 5. Cell type-specific responses to type I IFNs.**

(A) Venn diagram showing the number of genes significantly up-regulated in response to all tested type I IFNs in monocytes and lymphocytes. Gene lists are provided in Appendix Fig. S16. (B) Violin plots showing expression of selected genes in unstimulated ($n = 2690$ cells) or IFN-β-treated ($n = 3350$ cells) samples obtained from a single donor. (C) Heatmaps showing average expression of significantly up-regulated genes for each sample. Expression is represented as a z-score across all samples for each gene. Data are from one experiment with one donor. See also Appendix Figs. S16–S18.

to all type I IFNs. 229 genes were up-regulated by monocytes and 188 by lymphocytes, with 111 genes significantly up-regulated by both using padj < 0.05 and 1.5-fold induction as thresholds (Fig. 5A; Appendix Fig. S16). Genes specifically up-regulated by monocytes included *IFITM3*, *CXCL10* and *SIGLEC1* while genes specifically up-regulated by lymphocytes included *LAG3*, *ALOX5AP* and *CD48* (Fig. 5B,C). As expected, genes up-regulated by both monocytes and lymphocytes included those described as ISGs by many previous studies, such as *ISG15*, *IFI44* and *IFIT5* (Fig. 5B,C). Although these genes were selected based on their up-regulation by all five type I IFN subtypes tested, differences in the magnitude of the response were apparent, with IFN-β inducing many ISGs most strongly. This was in agreement with the mass cytometry and bulk RNAseq data indicating that different type I IFNs induce qualitatively similar responses. Subdivision of lymphocytes into T cells, B cells and NK cells revealed 50 genes and two lncRNAs only up-regulated in T cells, six genes and one lncRNA only up-regulated by B cells and two genes that were only up-regulated by NK cells (Appendix Fig. S17) using our significance and fold-change thresholds. Promoter analysis revealed no distinctive motifs associated with cell-type-specific ISGs, likely due in part to the limited number of genes in each category. Taken together, our scRNAseq analysis showed that transcriptional responses were qualitatively different between cell-types, including shared and cell-type specific ISGs.

In the bulk RNA-seq dataset, differences in the expression of genes induced by the different type I IFN subtypes were largely quantitative (Appendix Fig. S10). To investigate if this was also the case when cell type-specific ISGs were taken into account, we analysed the expression of genes induced only by one type I IFN subtype in monocytes and lymphocytes in all samples (Appendix Fig. S18). Consistent with our bulk RNAseq data, differences between different type I IFN subtypes were again quantitative rather than qualitative for most genes.

## ISG induction shows cell-to-cell heterogeneity

Next, we asked whether there was, for a given cell type, cell-to-cell variation in the transcriptional response to type I IFN stimulation. For example, some cells may be "super-responders" and others "non-responders". It is also possible that some cells up-regulate a distinct module of ISGs in response to type I IFN while others up-regulate a different set of ISGs, information that would be masked when averaging across the cell type cluster.

We chose to focus on classical monocytes as these represent a large cluster with many significantly up-regulated genes (i.e., 278, 330, 482, 643, 526 and 225 in response to IFN-α1, -α2a, -α10, -β, -ω and all five type I IFNs, respectively). Expression of each gene for each cell within this cell type was visualised using heatmaps and the genes clustered using Euclidean distance. This is shown for unstimulated and IFN-β-treated cells in Fig. 6A (see Appendix

Fig. S19 for this heatmap with gene name annotations for online viewing). This revealed that there was a large spectrum of ISG induction in IFN-β-stimulated cells, with some genes induced moderately (such as those in cluster 2, including classical ISGs such as *IFI44*, *OAS2*, *OAS3* and *OASL*), intermediately (clusters 4 and 6) and strongly (cluster 5). The genes in cluster 5 were all expressed at very low level in the unstimulated cells and therefore showed the greatest fold induction in expression upon stimulation with IFN. There were also differences in baseline expression of many of these ISGs, with most being not expressed/expressed at very low levels by most cells. Most of the 225 genes induced by all type I IFNs tested were up-regulated by all cells in the IFN-β-treated sample compared to the unstimulated cells. Interestingly, some genes displayed heterogeneous expression in the IFN-β-treated sample (Fig. 6A; Appendix Fig. S19) and in the samples stimulated with other type I IFNs (Appendix Fig. S20). These included *IFI27*, *CCL8, CCL2* and *CXCL10* (Appendix Fig. S19). To establish if expression of these four genes was linked, i.e. whether cells which up-regulated one of them also up-regulated the others, we clustered IFN-β stimulated classical monocytes based on their expression of these genes (Appendix Fig. S21). This revealed that while there were cells that up-regulated all four genes, there were also cells which expressed three or fewer, in all combinations, with no obvious patterns. This suggests it is unlikely that these four ISGs were induced in relation to each other.

We also visualised expression of this list of 225 genes up-regulated by classical monocytes in B cells and naive CD4 + T cells, two clusters with comparable numbers of cells (Appendix Fig. S22). As expected, a large number of these genes were unchanged in these cell types in response to type I IFN-treatment. Similarly, with the list of 94 genes up-regulated by B cells (Appendix Fig. S23) and 113 genes up-regulated by naive CD4 + T cells (Appendix Fig. S24), cell type-specific responses were clear. Genes that were up-regulated in all three cell types showed a similar degree of heterogeneity between individual cells (Appendix Fig. S25).

Finally, we calculated an ISG score for every cell in our dataset by summing the log-transformed expression values for each gene significantly up-regulated by the corresponding cell type. As shown for classical monocytes in Fig. 6B, the distribution of the scores for cells within a sample indicated that there are no "super-responder" cells and very few "non-responders". This was also the case for other cell types (Appendix Fig. S26). We also calculated a Core ISG score for each cell using the ten 'Core ISGs' (Fig. 6C; Appendix Fig. S15). This confirmed that all cells within a stimulated sample up-regulated expression of these genes, regardless of the type I IFN subtype used or cell type (Fig. 6C; Appendix Fig. S27). We therefore propose that an ISG score calculated from expression of these ten genes is sufficient to capture a transcriptional response to all type I IFN subtypes in all types of PBMCs.

Taken together, our single-cell transcriptomic analysis of type I IFN stimulated PBMCs identified ISGs induced in different white

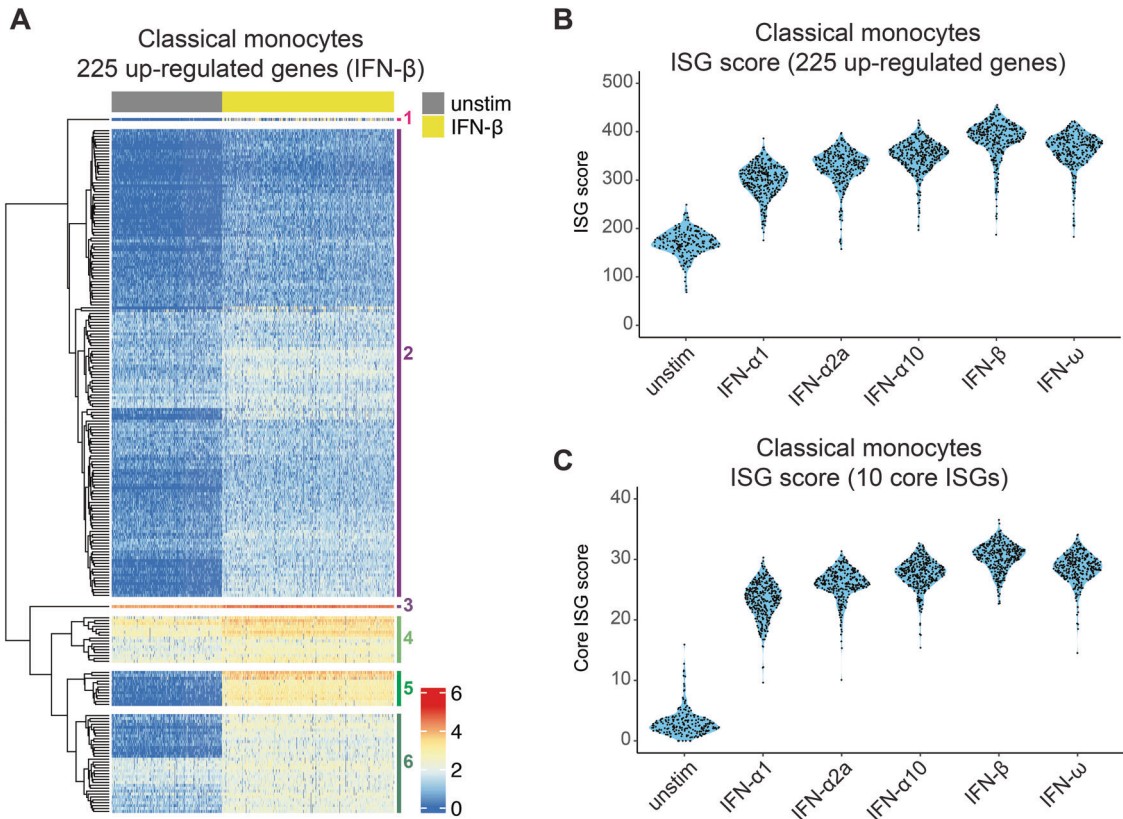

**Figure 6. Response of classical monocytes to type I IFNs.**

(A) Heatmap showing expression of the 225 genes significantly up-regulated in response to all tested type I IFNs in classical monocytes in the unstimulated and IFN-β-stimulated conditions. Each row represents a gene and each column a cell. An enlarged heatmap with annotated gene names is provided in Appendix Fig. S19. (B, C) Sinaplots with violin outline showing the ISG scores calculated using the 225 up-regulated genes in classical monocytes (B) and our list of ten core ISGs (C). ISG scores were determined on a per cell basis and are shown for classical monocytes with each black dot representing a single cell. Data are from one experiment with one donor. See also Appendix Figs. S19–S27.

blood cells. Some ISGs were induced broadly in all or many cell types, while others displayed cell-type-specific induction. In agreement with the mass cytometry and bulk RNAseq data, our scRNAseq analysis also suggested that different type I IFNs induced qualitatively similar responses.

## Discussion

Type I IFNs are a large family of cytokines, playing important roles in the immune system and in many diseases that range from infections to inflammatory and malignant conditions. Here, we employed single-cell technologies to generate comprehensive maps of signalling and transcriptomic changes in human white blood cells stimulated ex vivo with different type I IFNs. These data revealed responses shared between cell types as well as cell-type specific behaviours. Moreover, when comparing different type I IFN subtypes, we observed qualitatively similar responses that varied in magnitude.

An important technical consideration for our study was the normalisation of type I IFN doses used to treat cells (see also 'Limitations of the study' below). We relied on bioactivity (U/ml) that is measured by the manufacturer of recombinant type I IFNs

using a cytopathic effect (CPE) inhibition assay. In brief, the lung cancer cell line A549 is treated with type I IFN and is infected with the cytopathic encephalomyocarditis virus (EMCV). Control cells not treated with IFN are killed by EMCV, whereas cells treated with sufficient IFN survive. How, then, is it possible that different type I IFNs induce differing magnitudes of STAT phosphorylation and ISG expression despite being used at the same bioactivity? Cell survival in the CPE inhibition assay may be due to one or a few ISGs. Indeed, single ISGs can mediate powerful antiviral defence. For example, MX1 is crucial for host defence against influenza A virus (Grimm et al, 2007). Thus, similar bioactivity of different IFNs in A549 cells against EMCV-triggered cell death may not reflect the breadth of effects on many ISGs. Moreover, IFN-induced survival of A549 cells following EMCV infection is a binary readout. Induction of the relevant ISG(s) mediating protection beyond a threshold required for cell survival is unlikely to register in this assay. Thus, similar antiviral bioactivity (in the CPE inhibition assay) and differing magnitudes of modulation of ISG expression (at transcriptome level) are compatible.

Our datasets are freely available in multiple ways. Raw data can be accessed through repositories (see data availability statement). Lists of differentially expressed genes in different cell types in response to different type I IFNs are provided in the supplement to

this article. In addition, we provide our data via the Interactive SummarizedExperiment Explorer (iSEE) (Rue-Albrecht et al, 2018). This online tool [https://rehwinkellab.shinyapps.io/ifnresource/] is easy to use and requires no programming skills. Examples of its application include (1) querying a cell type of interest for genes induced or repressed by type I IFNs or for its signalling response and (2) analysing a gene (or signalling protein) of interest for baseline and type I IFN-regulated expression (or phosphorylation) in multiple populations of leukocytes. Another widely applicable result is the list of ten 'Core ISGs' induced in all types of PBMCs by all type I IFNs. For example, this set of ISGs will be useful as a biomarker to test for the presence of a type I IFN response in patients in whom the cellular basis of disease is unknown.

Monocytes were the cell type with the largest number of type I IFN-regulated genes, both in terms of the overall set of differentially expressed genes as well as genes affected only in monocytes but not in other cell types. This was likely due in part to the comparatively high baseline expression and/or phosphorylation of proteins involved in IFNAR signalling in monocytes (Appendix Fig. S2), bestowing an enhanced capacity to respond. Another important aspect likely determining which genes are regulated by type I IFN in different cell types is the epigenetic landscape. In the future, it would be interesting to investigate whether the contingent of ISGs in each cell type correlates with chromatin marks and accessibility using approaches such as ChIPseq and ATACseq.

In addition to protein-coding transcripts, we also provide a systematic analysis of polyadenylated ncRNAs using our bulk RNAseq data. NcRNAs are a heterogenous group of transcripts varying in length that do not encode proteins, including microRNA precursors, antisense and divergent transcripts, as well as lncRNAs. The expression of many ncRNAs is regulated and alterations to their expression can control cellular behaviour and be associated with human diseases. For example, several lncRNAs (Mattick and Rinn, 2015; Statello et al, 2021) are associated with the antiviral response and regulate the activity of pattern recognition receptors and signalling pathways leading to cytokine production, including the type I IFN signalling pathway (Ji et al, 2022; Suarez et al, 2020). These include NRIR (Negative Regulator of the Interferon Response) (Kambara et al, 2014), previously implicated in driving a type I IFN signature in monocytes in human autoimmune disease and viral infection (Bayyurt et al, 2021; Mariotti et al, 2019; Song et al, 2022). Indeed, NRIR was differentially expressed in our dataset (Appendix Fig. S12C,G). However, in contrast to NRIR, most ncRNAs regulated by type I IFNs in our dataset are uncharacterised and often unnamed (Appendix Fig. S12C). Our dataset thus provides a rich resource for future studies of these non-coding transcripts in the type I IFN response.

An important and open question in the type I IFN field pertains to the large number of genes encoding these cytokines. Many mammalian species, including bats, encode a multitude of type I IFNs (Hardy et al, 2004; Pavlovich et al, 2018; Zhou et al, 2016). This is indicative of evolutionary pressures leading to maintenance of many type I IFN genes. Simply put, why are there so many type I IFNs all using the same receptor?

One possible explanation is that different type I IFNs have unique activities against specific pathogens or in specific cell types (Buzzai et al, 2020; Foster et al, 2004; Harper et al, 2015; Hilkens et al, 2003; Karakoese et al, 2022; Lavender et al, 2016; Matos et al, 2019; Ortaldo et al, 1984; Rout et al, 2022; Schmitz et al, 2022;

Schuhenn et al, 2022; Tauzin et al, 2021; Xie et al, 2022). For example, IFN-α14 is particularly potent at inhibiting HIV-1 replication in different in vitro, ex vivo and in vivo models compared to IFN-α2 (Harper et al, 2015; Lavender et al, 2016; Rout et al, 2022; Tauzin et al, 2021). During SARS-CoV-2 infection, IFN-α5 exerts a strong antiviral effect in cultured human airway epithelial cells (Schuhenn et al, 2022). The latter observation correlates with induction of a subset of ISGs by IFN-α5 and other potently antiviral type I IFNs; these ISGs are not induced in this setting by other type I IFNs with poor anti-SARS-CoV-2 activity (Schuhenn et al, 2022). Moreover, clinical trials demonstrated distinct therapeutic effects. For example, IFN-α2 is effective in treating some viral diseases and some forms of cancer, whereas IFN-β is used to treat multiple sclerosis (Ioannou and Isenberg, 2000). We observed in PBMCs that different type I IFNs did not trigger qualitatively different cellular behaviours. Instead, the magnitude of the signalling and transcriptional responses varied between different type I IFNs. These divergent observations may be due to differences in haematopoietic and non-haematopoietic cells. For example, it is possible that in airway epithelial cells the stoichiometries and activities of proteins involved in IFNAR signalling are different to white blood cells and that this, perhaps in conjunction with alternate chromatin configurations, permits type I IFN-subtype-specific responses due to differences in their affinities for IFNAR. Studies comparing cells representing different tissues are therefore warranted, as is careful dissection of the responses to type I IFNs over varying doses and at different time points. It is tempting to speculate that additional factors such as type I IFN 'presentation' involving neighbouring cells in solid tissues or the soluble IFNAR2 isoform (Owczarek et al, 1997) in blood may play currently under-appreciated roles in controlling responses to type I IFNs.

Other possible explanations for the multitude of type I IFNs include pathogen encoded antagonists. Indeed, vaccinia virus encodes B18, a secreted protein that sequesters type I IFNs (Symons et al, 1995). Expansion of the type I IFN locus may have evolved to overcome the effects of B18 and potentially of similar, yet to be discovered, inhibitors, either simply by increasing gene dosage to saturate antagonists or more specifically through sequence diversification allowing escape of inhibition. Additionally, or alternatively, differences in the kinetics and signalling requirements for induction of type I IFNs may explain the large number of these cytokines. For example, the IFN-β promoter has elements recognised by multiple transcription factors including IRF3 and IRF7, NFκB and AP1 family members. In contrast, IFN-α genes are only controlled by IRF3 and IRF7 (Honda et al, 2006). Moreover, the different IFN-α genes are induced at different time points during viral infection (Marie et al, 1998). It is therefore possible that so many type I IFN genes have evolved to allow temporal, and perhaps also spatial and cell-type specific, control of their induction, including feedback loops. Indeed, IRF7 is encoded by an ISG and is an important part of a positive feed-forward circuit promoting strong IFN-α production by plasmacytoid dendritic cells (Izaguirre et al, 2003; Kerkmann et al, 2003). This scenario would be consistent with our findings that type I IFNs—once produced—induce qualitatively similar responses. We also note that quantitative differences, for example the particularly strong effects of IFN-β, may result in functionally distinct outcomes, including different clinical responses to treatment with different recombinant type I

IFNs (Ioannou and Isenberg, 2000). Clearly, understanding the diversity of type I IFNs remains an important challenge for future studies.

Taken together, we provide a data-rich and easily accessible resource of the responses of many different types of primary immune cells to type I IFNs. We anticipate that our datasets will instruct and inform research in many fields ranging from immunology and virology to cancer.

## Limitations of the study

This study was performed in PBMCs stimulated with type I IFNs ex vivo. Other cell types may show different behaviours in response to type I IFNs and additional factors such as the soluble form of IFNAR2 may impact in vivo responses. In-depth analysis of some cell types, such as rare dendritic cell subsets, was not possible due to the small number of cells. Enrichment of these cells prior to analysis will be necessary in future studies. Moreover, low per cell sequencing depth is a limitation of scRNAseq, likely resulting in underestimation of the number of ISGs, particularly of lowly expressed ISGs. Given that ncRNAs are generally expressed at lower levels than protein coding genes, drop-out rates were high in our scRNAseq dataset, precluding their analysis at single-cell level. We used recombinant type I IFNs for treatment. Doses were normalised using bioactivity (U/ml), determined by inhibition of the cytopathic effect of encephalomyocarditis virus in A549 cells. It is possible that the antiviral effect in this setting depends on induction of one or few ISGs, rather than on the breadth of effects of type I IFNs, potentially skewing normalisation. However, other ways of normalisation, for example using mass (pg/ml), have inherent limitations, too, because recombinant type I IFN preparations likely contain inactive protein.

## Methods

### Reagents and tools table

| Reagent/ Resource | Reference or Source | Identifier or Catalog Number |
|---|---|---|
| **Antibodies** | | |
| **Mass cytometry antibodies** | | |
| Anti-CD14 (TUK4)-Qdot655 | Thermo Fisher Scientific | Cat# Q10056, RRID:AB_2556446 |
| Anti-CD45(HI30)-89Y | Fluidigm | Cat# 3089003, RRID:AB_2661851 |
| Anti-CD11c(Bu15)-141Pr | BioLegend | Cat# 337221, RRID:AB_2562834 |
| Anti-CD11b(ICRF44)-142Nd | BioLegend | Cat# 301337, RRID:AB_2562811 |
| Anti-CD45RA(HI100)-143Nd | BioLegend | Cat# 304143, RRID:AB_2562822 |
| Anti-HLA-DR(L243)-144Nd | BioLegend | Cat# 307651, RRID:AB_2562826 |
| Anti-CD4(RPA-T4)-145Nd | BioLegend | Cat# 300541, RRID:AB_2562809 |

| Reagent/ Resource | Reference or Source | Identifier or Catalog Number |
|---|---|---|
| Anti-CD19(HIB19)-146Nd | BioLegend | Cat# 302247, RRID:AB_2562815 |
| Anti-CD20(2H7)-147Sm | BioLegend | Cat# 302343, RRID:AB_2562816 |
| Anti-CCR6(G034E3)-148Nd | BioLegend | Cat# 353427, RRID:AB_2563725 |
| Anti-CD56(NCA-M16.2)-149Sm | Fluidigm | Cat# 3149021B |
| Anti-pSTAT5(47)-150Nd | Fluidigm | Cat# 3150005 A, RRID:AB_2744690 |
| Anti-CD45RO(UCH-L1)-151Eu | BioLegend | Cat# 304239, RRID:AB_2563752 |
| Anti-CD27(O323)-152Sm | BioLegend | Cat# 302839, RRID:AB_2562817 |
| Anti-pSTAT1(4a)-153Eu | Fluidigm | Cat# 3153005A, RRID:AB_2744689 |
| Anti-CD1c(L161)-154Sm | BioLegend | Cat# 331502, RRID:AB_1088995 |
| Anti-CD123(6H6)-155Gd | BioLegend | Cat# 306027, RRID:AB_2562823 |
| Anti-p-p38(D3F9)-156Gd | Fluidigm | Cat# 3156002, RRID:AB_2661826 |
| Anti-pSTAT3(4/P-Stat3)-158Gd | Fluidigm | Cat# 3158005 A, RRID:AB_2811100 |
| Anti-pMAP-KAPK2(27B7)-159Tb | Fluidigm | Cat# 3159010, RRID:AB_2661828 |
| Anti-CD3(UCHT1)-160Gd | BioLegend | Cat# 300443, RRID:AB_2562808 |
| Anti-DNGR1(8F9)-161Dy | Fluidigm | Cat# 3161018B, RRID:AB_2810252 |
| Anti-IFNAR2(polyclonal)-162Dy | Abcam | Cat# ab56070, RRID:AB_880736 |
| Anti-STAT1(246523)-163Dy | Bio-Techne | Cat# MAB1490, RRID:AB_2159797 |
| Anti-IFNAR1(EP899Y)-164Dy | Abcam | Cat# ab213331 |
| Anti-CD161(HP-3G10)-165Ho | BioLegend | Cat# 339919, RRID:AB_2562836 |
| Anti-pNFkBp65(K10x)-166Er | Fluidigm | Cat# 3166006 A, RRID:AB_2847867 |
| Anti-CCR7(G043H7)-167Er | Fluidigm | Cat# 3167009 A, RRID:AB_2858236 |

| Reagent/ Resource | Reference or Source | Identifier or Catalog Number |
|---|---|---|
| Anti-pSTAT6(18/P-Stat6)-168Er | Fluidigm | Cat# 3168012 A, RRID:AB_2811103 |
| Anti-CD24(ML5)-169Tm | Fluidigm | Cat# 3169004B, RRID:AB_2688021 |
| Anti-CD141(M80)-170Er | BioLegend | Cat# 344102, RRID:AB_2201808 |
| Anti-pERK1/2(D13.14.4E)-171Yb | Fluidigm | Cat# 3171010 A, RRID:AB_2811250 |
| Anti-CD38(HIT2)-172Yb | Fluidigm | Cat# 3172007B, RRID:AB_2756288 |
| Anti-STAT3(124H6)-173Yb | Fluidigm | Cat# 3173003 A |
| Anti-pSTAT4(38/p-Stat4)-174Yb | Fluidigm | Cat# 3174005 A |
| Anti-CCR4(L291H4)-175Lu | Fluidigm | Cat# 3175035 A |
| Anti-CXCR3(G025-H7)-176Yb | BioLegend | Cat# 353733, RRID:AB_2563724 |
| Anti-CD8(RPA-T8)-198Pt | BioLegend | Cat# 301053, RRID:AB_2562810 |
| Anti-CD16(3G8)-209Bi | Fluidigm | Cat# 3209002B, RRID:AB_2756431 |
| **CITE-seq antibodies** | | |
| TotalSeq™-A0081 anti-human CD14 | BioLegend | Cat# 301855, RRID:AB_2734254 |
| TotalSeq™-A0391 anti-human CD45 | BioLegend | Cat# 304064, RRID:AB_2734266 |
| TotalSeq™-A0053 anti-human CD11c | BioLegend | Cat# 371519, RRID:AB_2749971 |
| TotalSeq™-A0161 anti-human CD11b | BioLegend | Cat# 301353, RRID:AB_2734249 |
| TotalSeq™-A0063 anti-human CD45RA | BioLegend | Cat# 304157, RRID:AB_2734267 |
| TotalSeq™-A0159 anti-human HLA-DR | BioLegend | Cat# 307659, RRID:AB_2750001 |
| TotalSeq™-A0072 anti-human CD4 | BioLegend | Cat# 300563, RRID:AB_2734247 |
| TotalSeq™-A0050 anti-human CD19 | BioLegend | Cat# 302259, RRID:AB_2734256 |
| TotalSeq™-A0100 anti-human CD20 | BioLegend | Cat# 302359, RRID:AB_2749984 |
| TotalSeq™-A0047 anti-human CD56 | BioLegend | Cat# 362557, RRID:AB_2749970 |

| Reagent/ Resource | Reference or Source | Identifier or Catalog Number |
|---|---|---|
| TotalSeq™-A0087 anti-human CD45RO | BioLegend | Cat# 304255, RRID:AB_2734268 |
| TotalSeq™-A0154 anti-human CD27 | BioLegend | Cat# 302847, RRID:AB_2750000 |
| TotalSeq™-A0160 anti-human CD1c | BioLegend | Cat# 331539, RRID:AB_2734326 |
| TotalSeq™-A0064 anti-human CD123 | BioLegend | Cat# 306037, RRID:AB_2749977 |
| TotalSeq™-A0034 anti-human CD3 | BioLegend | Cat# 300475, RRID:AB_2734246 |
| TotalSeq™-A0149 anti-human CD161 | BioLegend | Cat# 339945, RRID:AB_2749998 |
| TotalSeq™-A0071 anti-human CCR4 | BioLegend | Cat# 359423, RRID:AB_2749979 |
| TotalSeq™-A0140 anti-human CXCR3 | BioLegend | Cat# 353745, RRID:AB_2749993 |
| TotalSeq™-A0080 anti-human CD8a | BioLegend | Cat# 301067, RRID:AB_2734248 |
| TotalSeq™-A0083 anti-human CD16 | BioLegend | Cat# 302061, RRID:AB_2734255 |
| Purified anti-human DNGR1 | BioLegend | Cat# 353802, RRID:AB_10983070 |
| **Hashing antibodies** | | |
| TotalSeq™-A0251 anti-human Hashtag 1 | BioLegend | Cat# 394601, RRID:AB_2750015 |
| TotalSeq™-A0252 anti-human Hashtag 2 | BioLegend | Cat# 394603, RRID:AB_2750016 |
| TotalSeq™-A0253 anti-human Hashtag 3 | BioLegend | Cat# 394605, RRID:AB_2750017 |
| TotalSeq™-A0254 anti-human Hashtag 4 | BioLegend | Cat# 394607, RRID:AB_2750018 |
| TotalSeq™-A0255 anti-human Hashtag 5 | BioLegend | Cat# 394609, RRID:AB_2750019 |
| TotalSeq™-A0256 anti-human Hashtag 6 | BioLegend | Cat# 394611, RRID:AB_2750020 |
| **Oligonucleotides and other sequence-based reagents** | | |
| **CITE-seq: HTO sequences** | | |
| GTCAACTCTT-TAGCG; unstim | This study | n.a. |
| TGATGGCC-TATTGGG; ifna1 | This study | n.a. |
| TTCCGCCTCTC-TTTG; ifna2a | This study | n.a. |

| Reagent/Resource | Reference or Source | Identifier or Catalog Number |
|---|---|---|
| AGTAAGTT-CAGCGTA; ifna10 | This study | n.a. |
| AAG-TATCGTTTCGC-A; ifnb | This study | n.a. |
| GGTTGCCA-GATGTCA; ifno | This study | n.a. |
| **CITE-seq: ADT sequences** | | |
| CAT-GATTGGCTC; DNGR1 | This study | n.a. |
| TCTCA-GACCTCCGTA; CD14 | This study | n.a. |
| TGCAAT-TACCCGGAT; CD45 | This study | n.a. |
| TACGCCTA-TAACTTG; CD11c | This study | n.a. |
| GACAAGT-GATCTGCA; CD11b | This study | n.a. |
| TCAATCCTTCC-GCTT; CD45RA | This study | n.a. |
| AATAGCGAG-CAAGTA; HLADR | This study | n.a. |
| TGTTCCCGCTC-AACT; CD4 | This study | n.a. |
| CTGGGCAAT-TACTCG; CD19 | This study | n.a. |
| TTCTGGGTCCC-TAGA; CD20 | This study | n.a. |
| TCCTTTCCTGA-TAGG; CD56 | This study | n.a. |
| CTCCGAAT-CATGTTG; CD45RO | This study | n.a. |
| GCACTCCTG-CATGTA; CD27 | This study | n.a. |
| GAGCTACTT-CACTCG; CD1c | This study | n.a. |
| CTTCACTCTGT-CAGG; CD123 | This study | n.a. |
| CTCATTG-TAACTCCT; CD3 | This study | n.a. |
| GTACG-CAGTCCTTCT; CD161 | This study | n.a. |
| AGCTTACCTG-CACGA; CCR4 | This study | n.a. |
| GCGATGGTA-GATTAT; CXCR3 | This study | n.a. |
| GCTGCGCTTTC-CATT; CD8a | This study | n.a. |

| Reagent/Resource | Reference or Source | Identifier or Catalog Number |
|---|---|---|
| AAGTT-CACTCTTTGC; CD161 | This study | n.a. |
| **Oligos for ADT and HTO library amplification** | | |
| ADT CITE-seq cDNA PCR additive primer: CCTTGGCACC CGAGAATTCC | Biomers.net | Stoeckius et al, 2018 and https://citeseq. files.wordpress.com/ 2019/02/cite-seq_ and_hashing_ protocol_190213.pdf |
| HTO Cell Hashing cDNA PCR additive primer: GTGACTG-GAGTTCA-GACGTGTGCTC | Biomers.net | Stoeckius et al, 2018 and https://citeseq. files.wordpress.com/ 2019/02/cite-seq_ and_hashing_ protocol_190213.pdf |
| Illumina Small RNA RPI1 primer: CAAGCAGAA-GACGGCATAC-GAGATCGT-GATGTGACTG-GAGTTCCTTGG-CACCCGA-GAATTCCA | Biomers.net | Stoeckius et al, 2018 and https://citeseq. files.wordpress.com/ 2019/02/cite-seq_ and_hashing_ protocol_190213.pdf |
| Illumina Small RNA RPI2 primer: CAAGCAGAA-GACGGCATAC-GAGATA-CATCGGT-GACTG-GAGTTCCTTGG-CACCCGA-GAATTCCA | Biomers.net | Stoeckius et al, 2018 and https://citeseq. files.wordpress.com/ 2019/02/cite-seq_ and_hashing_ protocol_190213.pdf |
| Illumina TruSeq D701_s primer: CAAGCAGAA-GACGGCATAC-GAGATCGAG-TAATGTGACTG-GAGTTCA-GACGTGTGC | Biomers.net | Stoeckius et al, 2018 and https://citeseq. files.wordpress.com/ 2019/02/cite-seq_ and_hashing_ protocol_190213.pdf |
| Illumina TruSeq D702_s primer: CAAGCAGAA-GACGGCATAC-GAGATTCTCCG-GAGTGACTG-GAGTTCA-GACGTGTGC | Biomers.net | Stoeckius et al, 2018 and https://citeseq. files.wordpress.com/ 2019/02/cite-seq_ and_hashing_ protocol_190213.pdf |
| **Chemicals, Enzymes and other reagents** | | |
| **Recombinant human type I IFNs** | | |
| IFN-α1 | PBL Assay Science | Cat# 11125-1 |
| IFN-α2a | PBL Assay Science | Cat# 11100-1 |
| IFN-α2b | PBL Assay Science | Cat# 11105-1 |
| IFN-α4 | PBL Assay Science | Cat# 11180-1 |
| IFN-α5 | PBL Assay Science | Cat# 11135-1 |
| IFN-α6 | PBL Assay Science | Cat# 11165-1 |
| IFN-α7 | PBL Assay Science | Cat# 11160-1 |
| IFN-α8 | PBL Assay Science | Cat# 11115-1 |

| Reagent/Resource | Reference or Source | Identifier or Catalog Number |
|---|---|---|
| IFN-α10 | PBL Assay Science | Cat# 11120-1 |
| IFN-α14 | PBL Assay Science | Cat# 11145-1 |
| IFN-α16 | PBL Assay Science | Cat# 11190-1 |
| IFN-α17 | PBL Assay Science | Cat# 11150-1 |
| IFN-α21 | PBL Assay Science | Cat# 11130-1 |
| IFN-β | PBL Assay Science | Cat# 11415-1 |
| IFN-ω | PBL Assay Science | Cat# 11395-1 |
| **Cell culture reagents** | | |
| Lymphoprep™ | Stemcell Technologies | Cat# 07851 |
| RPMI-1640 Medium | Sigma-Aldrich | Cat# R8758 |
| Fetal Calf Serum | | |
| **Mass cytometry reagents** | | |
| Cell-ID 20-Plex Pd Barcoding Kit | Fluidigm | Cat# 201060 |
| Cell-ID Cisplatin | Fluidigm | Cat# 201064 |
| Cell-ID Intercalator-Ir | Fluidigm | Cat# 201192B |
| EQ Four Element Calibration Beads | Fluidigm | Cat# 201078 |
| Maxpar Water | Fluidigm | Cat# 201069 |
| 16% Paraformaldehyde solution | Thermo Fisher | Cat# 28906 |
| Maxpar X8 Antibody Labeling Kit | Fluidigm | |
| Bond-Breaker® TCEP Solution | Thermo Fisher Scientific | Cat# PI77720 |
| Antibody Stabilisation Buffer | Candor | Cat# 131050 |
| Human TruStain FcX (FcR Blocking Solution) | BioLegend | Cat# 422301 |
| Maxpar Cell Staining Buffer | Fluidigm | Cat# 201068 |
| Maxpar Fix and Perm Buffer | Fluidigm | Cat# 201067 |
| Maxpar PBS | Fluidigm | Cat# 201058 |
| Maxpar Fix I Buffer (5X) | Fluidigm | Cat# 201065 |
| Maxpar Perm Buffer (10X) | Fluidigm | Cat# 201057 |
| OneComp eBeads Compensation Beads | Invitrogen | Cat# 01-1111-42 |
| **Bulk RNAseq reagents** | | |
| RNeasy Mini Plus Kit | Qiagen | Cat# 74136 |
| Qubit RNA BR Assay Kit (used with Qubit 3.0) | Thermo Fisher Scientific | Cat# Q10210 |

| Reagent/Resource | Reference or Source | Identifier or Catalog Number |
|---|---|---|
| **Taqman assays** | | |
| HPRT | Thermo Fisher Scientific | Cat# Hs99999909_m1 |
| IFIT3 | Thermo Fisher Scientific | Cat# Hs01922752_s1 |
| IFIT1 | Thermo Fisher Scientific | Cat# Hs01911452_s1 |
| MX1 | Thermo Fisher Scientific | Cat# Hs00895608_m1 |
| ISG15 | Thermo Fisher Scientific | Cat# Hs01921425_s1 |
| IFI44 | Thermo Fisher Scientific | Cat# Hs00951349_m1 |
| IFI6 | Thermo Fisher Scientific | Cat# Hs00242571_m1 |
| IFI44L | Thermo Fisher Scientific | Cat# Hs00915292_m1 |
| **CITEseq reagents** | | |
| DPBS | Thermo Fisher Scientific | Cat# 14190144 |
| Bovine Serum Albumin | Sigma-Aldrich | Cat# A7906 |
| Triton X-100 | Promega | Cat# H5141 |
| Human TruStain FcX blocking solution | Biolegend | Cat# #422301 |
| Chromium Single Cell 3' GEM, Library & Gel Bead Kit v3 | 10X Genomics | Cat# PN-1000092 |
| Chromium Single Cell B Chip Kit | 10X Genomics | Cat# PN-1000154 |
| Chromium i7 Multiplex Kit | 10X Genomics | Cat# PN-120262 |
| Low TE Buffer | Thermo Fisher Scientific | Cat# 12090-015 |
| SPRIselect Reagent Kit | Beckman Coulter | Cat# B23318 |
| Glycerin, 50% (v/v) Aqueous Solution | Ricca Chemical Company | Cat# 3290-32 |
| Qiagen Buffer EB | Qiagen | Cat# 19086 |
| High Sensitivity DNA Kit (used with Agilent Bioanalyzer 2100) | Agilent | Cat# 5067-4626 |
| High Sensitivity D5000 ScreenTape (used with Tapestation 4200) | Agilent | Cat# 5067-5584 |
| Qubit dsDNA HS Assay Kit | Thermo Fisher Scientific | Cat# Q332854 |
| KAPA Library Quantification Kit for Illumina Platforms | KAPA Biosystems | Cat# KK4824 |

| Reagent/ Resource | Reference or Source | Identifier or Catalog Number |
|---|---|---|
| Nextseq 500/ 550 Hi Output kit v2.5 (150 cycles) | Illumina | Cat# 20024907 |
| **Software** | | |
| Helios CyTOF Software v6.7 | Fluidigm | https://www. standardbio.com/ |
| CATALYST (v1.5.3.23) | Chevrier et al, 2018, Cell Systems | http://bioconductor. org/packages/ release/bioc/html/ CATALYST.html |
| cytofCore (v0.4) | Bruggner et al (2023) | https://github.com/ nolanlab/cytofCore/ |
| FlowJo v10.8 | FlowJo | https://www.flowjo. com/ |
| CATALYST (v1.18.1) | Crowell et al (2022) | https://github.com/ HelenaLC/ CATALYST |
| QuantStudio 7-flex PCR software | Thermo Fisher Scientific | https://www. thermofisher.com/ us/en/home/global/ forms/life-science/ quantstudio-6-7-flex-software.html |
| GraphPad Prism v9.4.0 | GraphPad | https://www. graphpad.com/ |
| FastQC v0.11.9 | Andrews S (2010) | https://www. bioinformatics. babraham.ac.uk/ projects/fastqc/ |
| Kallisto v0.46.1 | Bray et al (2016) Nature Biotechnology | https://pachterlab. github.io/kallisto/ |
| tximport v1.26.1 | Soneson et al (2015) F1000Research | https://bioconductor. org/packages/ release/bioc/html/ tximport.html |
| AnnotationHub v3.6.0 | Morgan and Shepherd (2022). AnnotationHub: Client to access AnnotationHub resources | https://bioconductor. org/packages/ release/bioc/html/ AnnotationHub.html |
| DESeq2 v1.36.1 | Love et al (2014) Genome Biology | https://bioconductor. org/packages/ release/bioc/html/ DESeq2.html |
| goseq v1.50.0 | Young et al (2010) Genome Biology | https://bioconductor. org/packages/ release/bioc/html/ goseq.html |
| pheatmap v1.0.12 | Kolde (2019) Pheatmap: Pretty Heatmaps | https://cran.r-project.org/web/ packages/pheatmap/ index.html |
| ComplexHeatmap v2.14.0 | Gu et al (2016) Bioinformatics | https://bioconductor. org/packages/ release/bioc/html/ ComplexHeatmap. html |

| Reagent/ Resource | Reference or Source | Identifier or Catalog Number |
|---|---|---|
| EnhancedVolcano v1.16.0 | Blighe et al, (2018). EnhancedVolcano: Publication-ready volcano plots with enhanced colouring and labeling | https://github.com/ kevinblighe/ EnhancedVolcano |
| Cell Ranger v3.0.2 | 10X Genomics | https://www. 10xgenomics.com/ |
| bcl2fastq v2.20.0.422 | Illumina | https://www. illumina.com/ |
| CITE-Seq-Count v1.4.2 | Github | https://github.com/ Hoohm/CITE-seq-Count |
| Seurat v4.1.1 | Butler et al (2018) Nature Biotechnology | https://github.com/ satijalab/seurat |
| scCustomize v1.1.1 | Marsh SE (2021). scCustomize: custom visualizations & functions for streamlined analyses of single cell sequencing | https://github.com/ samuel-marsh/ scCustomize |
| eulerr v7.0.0 | Larsson J (2022). eulerr: Area-Proportional Euler and Venn Diagrams with Ellipses | https://cran.r-project.org/web/ packages/eulerr/ index.html |
| ggforce v0.4.1.9000 | Pedersen T (2022). ggforce: Accelerating 'ggplot2' | https://cran.r-project.org/web/ packages/ggforce/ index.html |
| tidyverse v1.3.2 | Wickham et al (2019) Journal of Open Source Software | https://www. tidyverse.org/ |

## Study subjects

PBMCs were isolated from the peripheral blood of healthy, anonymous donors using Lymphoprep (Stemcell Technologies), according to the manufacturer's instructions. Ethics oversight: This work was carried out in accordance with the EU Directive 2004/23/ EC and the UK Human Tissue Act 2004 (HTA). The Weatherall Institute of Molecular Medicine (WIMM) holds a HTA licence (number 12433).

## Data acquisition

### Mass cytometry (CyTOF)

PBMCs were washed in serum-free RPMI and then resuspended at $10^7$ cells/ml in serum-free RPMI containing 0.5 mM Cell-ID Cisplatin (Fluidigm) and incubated at 37 °C for 5 min. Cells were washed with RPMI containing 10% (v/v) FCS (Sigma) and 2 mM L-Glutamine (R10), centrifuged at $300 \times g$ for 5 min before being resuspended to $6 \times 10^7$ cells/ml in R10 and rested at 37 °C for 15 min. 50 μl of cells ($3 \times 10^6$) cells were transferred to 15 ml falcon

tubes for stimulation and antibody staining. Antibody staining was performed as described previously (Nienaltowski et al, 2021). Antibodies are listed in the Reagents and Tools table. Staining for CD14, CCR6, CD56, CD45RO, CD27, CCR7, CCR4 and CXCR3 was done for 30 min in R10 at 37 °C, prior to stimulation/fixation as epitopes recognised by these antibodies are sensitive to fixation. Cells were stimulated with the indicated concentration of type I IFN (Dataset EV2) diluted in R10 for the indicated time at 37 °C. After washing with 5 ml Maxpar PBS (Fluidigm), cells were fixed with 1 X Maxpar Fix I Buffer (Fluidigm) for 10 min at room temperature before being washed with 1.5 ml Maxpar Cell Staining Buffer (CSB, Fluidigm). All centrifugation steps after this point were at $800 \times g$ for 5 min. Cells were barcoded using Cell-ID 20-Plex Pd Barcoding Kit (Fluidigm), according to the manufacturer's instructions, and washed twice with CSB before samples were pooled and counted. All further steps were performed on pooled cells. Fc receptors were blocked using Fc Receptor Binding Inhibitor Antibody (eBioscience), diluted 1:10 in CSB for 10 min at room temperature. Surface antibody staining mixture was added directly to the blocking solution and incubated for 30 min at room temperature. Cells were washed twice with CSB, resuspended in ice-cold methanol and stored at −80 °C overnight. After washing twice with CSB, cells were stained with intracellular antibody staining mixture for 30 min at room temperature before two further washes in CSB. Cells were resuspended in 1.6% (v/v) formaldehyde (Pierce, #28906) diluted in Maxpar PBS and incubated for 10 min at room temperature. Cells were resuspended in 125 mM cell-ID Intercalator (Fluidigm) diluted in Maxpar Fix and Perm Buffer (Fluidigm) and incubated overnight at 4 °C. Compensation beads (OneComp eBeads Compensation Beads, Invitrogen, #01-111-42) stained with 1 µl of each antibody were also prepared. The following day, cells and compensation beads were washed twice with CSB and twice with Maxpar water (Fluidigm), mixed with a 1:10 volume of EQ Four Element Calibration Beads (Fluidigm) before acquisition on a Helios Mass Cytometer (Fludigm) using the HT injector.

### qRT-PCR

Freshly isolated PBMCs were stimulated with 250 U/ml of IFN-α1 or IFN-β in R10 or left unstimulated ($3 \times 10^6$ cells/sample) in a volume of 2 ml in a non-tissue culture treated 12-well plate and incubated for the time indicated in the figure legend at 37 °C. RNA was extracted using a RNeasy Mini Plus Kit including a gDNA eliminator column step, according to the manufacturer's instructions. cDNA synthesis was performed with SuperScript IV reverse transcriptase (Thermo Fisher Scientific) and oligo(dT)$_{12-18}$ primers (Thermo Fisher Scientific). 15 ng of cDNA was amplified using Taqman Universal PCR Mix (Thermo Fisher Scientific) and Taqman probes (see Reagents and Tools table) in 5 µl reactions. qPCR was performed on a QuantStudio 7 Flex real-time PCR system (Applied Biosystems). mRNA expression data were normalised to *HPRT* and analysed by the comparative $C_T$ method. Fold changes in expression from unstimulated samples for each time point were calculated for the IFN-treated samples.

### Bulk RNA-seq

Freshly isolated PBMCs were stimulated with 250 U/ml of each type I IFN in R10 ($3 \times 10^6$ cells/sample) in a volume of 2 ml in a non-tissue culture treated 12-well plate and incubated for 24 h at 37 °C. Cells were harvested by gentle pipetting and centrifuged at

$300 \times g$ for 5 min. The pellet was resuspended in 350 µl RLT buffer from an RNeasy Plus Mini Kit (QIAGEN) and transferred to a QiaShredder column. After homogenisation by centrifuging at $16,000 \times g$ for 2 min, RNA was extracted according to the manufacturer's instructions, including a gDNA eliminator column step. RNA was quantified using a Qubit 3.0 RNA BR (broad-range) Assay Kit (Invitrogen) and using RiboGren (Invitrogen) on a FLUOstar OPTIMA plate reader (BMG Labtech). The RNA size profile and integrity was assessed using a Tapestation 4200 with High Sensitivity RNA ScreenTape (Agilent Technologies). Input material was normalised to 100 ng prior to library preparation by the Oxford Genomics Centre. Polyadenylated transcript enrichment and strand specific library preparation was completed using a NEBNext Ultra II mRNA kit (New England Biolabs) following the manufacturer's instructions. Libraries were amplified (17 cycles) using in-house unique dual indexing primers (Lamble et al, 2013). Individual libraries were normalised using a Qubit, and the size profile was analysed using a Tapestation before individual libraries were pooled. The pooled library was diluted to ~10 nM, denatured and further diluted prior paired-end sequencing (150 bp reads) on a NovaSeq 6000 platform (Illumina, NovaSeq 6000 S2/S4 reagent kit, 300 cycles), yielding between 38 and 55 million reads/sample.

### scRNAseq: cell stimulation and capture

Freshly isolated PBMCs were stimulated with 250 U/ml of each type I IFN in R10 ($3 \times 10^6$ cells/sample) in a volume of 2 ml in a non-tissue culture treated 12-well plate and incubated for 24 h at 37 °C. Cells were harvested by gentle pipetting, centrifuged at $300 \times g$ for 5 min, resuspended in 1 ml R10 and the total cell number was determined. $0.5 \times 10^6$ cells were transferred to a low-binding 1.5 ml tube and centrifuged at $300 \times g$ for 5 min at 4 °C. Cell pellets were resuspended in 100 µl of staining buffer (2% (v/v) BSA/0.01% (v/v) Tween-20 in cold RNase-free PBS + 10 µl Human TruStain FcX blocking solution (#422301, BioLegend)) and incubated for 10 min at 4 °C. A mastermix of CITEseq antibodies (ADTs) was prepared consisting of 0.5 µg of each antibody and added to each sample together with 0.5 µg of unique Cell Hashing antibody (HTOs) (Reagents and Tools table) and cells were stained for 30 min at 4 °C. Cells were washed three times with 1 ml staining buffer, centrifuging at $350 \times g$ for 5 min at 4 °C. Cells were passed through a 35 µm cell strainer (Falcon) prior to the final wash step and were then resuspended in 200 µl staining buffer and counted. $1 \times 10^5$ cells from each sample were pooled and passed over a 40 µm cell strainer (Falcon). After centrifugation at $350 \times g$ for 5 min at 4 °C, the cell pellet was resuspended in 200 ml cold RNase-free PBS to give a concentration of 1500 cells/µl. Two lanes of a Chromium Single Cell Chip B (10X Genomics) were each superloaded with 30,000 cells, with a target of 15,000 single cells per lane (Stoeckius et al, 2018). Generation of gel beads in emulsion (GEMs), GEM-reverse transcription, clean up and cDNA amplification were performed using a Chromium Single Cell 3′ Reagent Kit (v3), according to the manufacturer's instructions, with the addition of 2 pmol each of ADT and HTO additive primers at the cDNA amplification step (Stoeckius et al, 2018). During cDNA cleanup, the ADT- and HTO-containing supernatant fraction was separated from the cDNA fraction derived from cellular mRNAs using 0.6X SPRI beads (Beckman Coulter).

### scRNAseq: library generation and sequencing

The mRNA-derived cDNA library was prepared according to the standard Chromium Single Cell 3′ Reagent Kit (v3). ADTs and

HTOs were purified using two 2X SPRI selection according to Stoeckius et al (2018) and amplified using separate PCR reactions to generate the ADT and HTO libraries with ten cycles of amplification. PCR products were purified using 1.6X SPRI purification. Quality control of libraries was performed using a Qubit 3.0 dsDNA HS (high sensitivity) Assay Kit (Invitrogen) and BioAnalyzer High Sensitivity DNA Chip (Agilent). Libraries were diluted to 2 nM and pooled for sequencing in the following proportions: 80% cDNA, 10% ADT, 10% HTO. Libraries were sequenced twice on a NextSeq 500 with 150 bp paired-end reads (Illumina). Sequencing runs were pooled, yielding a total of $5 \times 10^8$ reads for each 10X lane ($1 \times 10^9$ reads in total).

## Data analysis

### Mass cytometry (CyTOF)

Data were normalised, randomised and concatenated using Helios CyTOF Software (v6.7) (Fluidigm). Compensation and debarcoding were performed in R (v3.5.1) using CATALYST (v1.5.3.23) (Chevrier et al, 2018). FCS files were rewritten using the updatePanel function of cytofCore (v0.4). Intact, single, live, CD45+ cells were manually gated using FlowJo (v10.8) and exported as new FCS files. Data were transformed and cells clustered using FlowSOM as described in cytofWorkflow (Nowicka et al, 2017) using CATALYST (v1.18.1) in R (4.1.3). Cell populations were manually identified by expression of known phenotyping markers and clusters merged and annotated.

### Bulk RNAseq

Sequencing data were processed using an optimised fork of the community-curated Nextflow nf-core/rnaseq pipeline (v3.18.0) (da Veiga Leprevost et al, 2017; Di Tommaso et al, 2017; Ewels et al, 2020; Gruning et al, 2018). Briefly, raw reads were trimmed with fastp (v0.23.4) (Chen et al, 2018) to remove sequencing adaptors, low-quality bases and poly-A/G tails. The quality of the trimmed reads was assessed using FastQC (v0.12.1, bioinformatics.babraham.ac.uk/projects/fastqc). Trimmed reads were aligned to the GRCh38 human genome assembly with STAR (v2.7.11b) (Dobin et al, 2013), using the full GENCODE annotation (v48) (Mudge et al, 2025) and excluding only automatically annotated loci (annotation level 3). To improve the accuracy of abundance quantification, the annotation was further filtered to retain only RNA models that were either considered high-confidence in GENCODE (i.e., MANE Select or transcript support level (TSL) 1 or 2), or had all splice junctions supported by at least 5 uniquely or 20 non-uniquely mapped reads across all samples. These filtered models were used as input for abundance estimation with Salmon (v1.10.3) (Patro et al, 2017). To account for technically indistinguishable isoforms, transcripts were collapsed into groups using Terminus (v0.1.0) (Sarkar et al, 2020). The resulting per-transcript-group expression values were imported into DESeq2 (v1.46.0) (Love et al, 2014; Soneson et al, 2015) for differential expression analysis using a multi-factor design formula to control for donor-specific effects. Statistical testing was restricted to transcript groups in which all samples from at least one condition had a minimum of 20 assigned reads, ensuring the focus on robustly expressed transcription units. Estimated fold changes were further shrunk using apeglm (v1.28.0) (Zhu et al, 2019), and p-values were adjusted for multiple testing using the Benjamini-Hochberg

method, applied separately to coding and non-coding groups. Differentially expressed transcript groups (DETs) were defined as those with an adjusted p-value ≤ 0.05 and an absolute $\log_2$(fold change) ≥ 0.585. Gene ontology (GO) analysis was performed using GSEApy (v1.1.9) (Fang et al, 2023), using all expressed genes as a background.

### scRNAseq

BCL files were demultiplexed and converted to Fastq files using the mkfastq pipeline of Cell Ranger (v3.0.2) and bcl2fastq (v2.20.0.422) and read quality assessed using FastQC (v0.11.9). Raw sequencing reads for the cDNA libraries were processed into count matrices using cellranger count (Cell Ranger v3.0.2) and aligned to Human reference 3.0.0 (GRCh38) (10X Genomics), which includes annotation of protein-coding genes, lncRNAs, antisense transcripts, immunoglobulin genes/pseudogenes and T cell receptor genes. Raw sequencing reads from the ADT and HTO libraries were processed into count matrices using CITE-Seq-Count (v1.4.2) (Roelli et al, 2019). The count matrices for cDNA, ADT and HTO libraries were imported into R (v4.1.3) and analysed using Seurat (v 4.1.1). Cells with <200 or >5000 genes or >15% of mitochondrial reads were excluded. The HTO data was normalised using centred log-ratio (CLR) transformation and samples demultiplexed using the HTODemux function of Seurat. Count data for the RNA assay was normalised using sctransform v2, regressing out gene expression from ribosomal proteins and returning all genes. ADT data was normalised using CLR. Data from the two 10X lanes were merged and data scaled. The multimodal object was split into one object per sample and variable features normalised and identified using sctransform v2, using method "glGamPoi". PrepSCTIntegration was performed on the six objects using all genes as anchor features and PCA performed. Integration anchors were identified and used to create an integrated data assay normalised using sctransform. Dimensionality reduction was performed using Principal Component Analysis (PCA) using all features (genes) for the RNA assay and all features (antibodies) for the ADT assay. Weighted Nearest Neighbour (WNN) analysis (Hao et al, 2021) was used to improve clustering by combining the RNA and ADT data. FindMultiModalNeighbors was performed using 30 dimensions for RNA and 18 for ADT and the data visualised using UMAP. Clusters were determined using FindClusters using SLM algorithm with a resolution of 0.8 and cell types identified using expression of mRNA and protein markers.

Differential expression analysis was performed on the RNA assay which was first log normalised. The integrated object was subset by cell type and FindMarkers run for each type I IFN versus the unstimulated sample using a Wilcoxon rank-sum test with a logfc threshold of 0.25 and genes that are detected in a minimum of 10% of cells. DEGs were defined as having an adjusted p value of <0.05. Seurat (v4.1.1), Pheatmap (v1.0.12), ComplexHeatmap (v2.14.0), scCustomize (v1.1.1), eulerr (v7.0.0) and ggforce (v0.4.1.9000) were used for data visualisation.

### Promoter motifs

All transcripts from GENCODE, excluding those annotated at level 3, were matched to promoter-like sequences (PLSs) from the ENCODE Registry of cis-Regulatory Elements (V3) (Consortium et al, 2020), allowing up to 500 bp between a PLS and an annotated transcription start site (TSS). When multiple PLSs were associated

with one or more TSSs, all were retained for further analysis. Matched regions were uniformly extended to 350 bp, and for transcripts without a matched PLS, promoters were imputed as 350 bp sequences centred on the TSS.

De novo motif discovery was performed using STREME (v5.5.5) (Bailey, 2021) with default parameters, holding out 25% of sequences for the test set. Specifically, we compared transcript groups up- or down-regulated in bulk RNAseq following stimulation with a given IFN to a background set of transcripts that were consistently expressed across all conditions (TPM ≥ 10) and not differentially expressed in response to any IFN. Additional comparisons included transcripts differentially regulated by distinct IFNs, as well as promoters of transcripts corresponding to ISGs exhibiting cell type specificity in scRNAseq. For each comparison, positive and negative sets consisted of all promoters associated with the relevant transcript groups (or genes, in the case of scRNAseq), excluding any overlapping regions shared by both sets. Only motifs that were statistically significant (E-value ≤ 1e−2) and strongly enriched relative to the background (enrichment ≥ 4.0) were considered representative.

To correlate the binding potential of known TFs with observed expression changes in bulk RNAseq, we analysed promoters using the JASPAR 2024 Vertebrata CORE database (Rauluseviciute et al, 2024). First, we scored each promoter with every TF position weight matrix (PWM), taking the maximum score from the forward or reverse complement strand. These raw scores were then Z-normalised using a mean and standard deviation derived from promoters matched to high-quality transcripts (MANE Select or TSL 1 or 2). To account for motif redundancy, we summed the normalised scores of all PWMs within JASPAR-defined TF clusters and Z-scored this sum again to generate a final, per-promoter cluster score. Each transcript group was then assigned the maximum cluster activity score from all of its associated promoters. Next, we tested the association between these scores and IFN-induced expression changes. For each TF cluster, we partitioned transcript groups into a "likely regulated" set (Z-score ≥ 2) and a "likely unregulated" set (Z-score ≤ −1). For each differential expression comparison, we then compared the fold-change values between these two sets using a Mann–Whitney U test, considering only transcript groups with robust expression (median TPM ≥ 10) in either mock or treated cells. This process was repeated for every motif cluster across all treatments (i.e., individual IFNs vs. mock), and the resulting p-values were adjusted for multiple testing using the Benjamini-Hochberg correction.

## Data availability

All data are available in the manuscript and associated supplementary files and at https://rehwinkellab.shinyapps.io/ifnresource/. Mass cytometry data generated during this study has been deposited at Flow Repository (FR-FCM-Z655, http://flowrepository.org/id/FR-FCM-Z655) and is publicly available. The data used in Fig. 1 was published previously (Data ref: Nienaltowski et al, 2021). Sequencing data generated during this study has been deposited on ENA and is publicly available (PRJEB60774). Original code is publicly available (https://github.com/alnfedorov/ifn-resource) or from the corresponding author upon request.

The source data of this paper are collected in the following database record: biostudies:S-SCDT-10_1038-S44319-026-00750-3.

## Peer review information

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

## Acknowledgements

The authors would like to thank Clare Hardman and Yi-Ling Chen (MRC HIU) for assistance with phlebotomy, and Hayley Evans and David Fawkner-Corbett (MRC HIU) for technical advice. We would like to acknowledge Michalina Mazurczyk and Julia Menzies in the MRC WIMM Mass Cytometry Facility for providing technical expertise. The facility was supported by the MRC HIU core funded project (MC_UU_00008) and the Oxford Single Cell Biology Consortium (MR/M00919X/1). We thank Neil Ashley in the MRC WIMM Single Cell Core Facility for his help with the CITE-seq experiments. The facility was supported by the MRC MHU (MC_UU_12009), the Oxford Single Cell Biology Consortium (MR/M00919X/1), the WT-ISSF (097813/Z/11/B#) and WIMM Strategic Alliance awards G0902418 and MC_UU_12025. We would like to acknowledge Tim Rostron in the Sequencing facility at the MRC WIMM for assistance with sequencing services. The facility was supported by the MRC HIU (MC_UU_00008) and by the EPA fund (CF268). AF acknowledges funding from the Clarendon Fund. This work was funded by the UK Medical Research Council [MRC core funding of the MRC Human Immunology Unit; JR; MC_UU_00008]. The funders had no role in study design, data collection and analysis, decision to publish, or preparation of the manuscript.

## Author contributions

**Rachel E Rigby**: Conceptualization; Data curation; Software; Formal analysis; Validation; Investigation; Visualization; Methodology; Writing—original draft; Project administration; Writing—review and editing. **Kevin Rue-Albrecht**: Data curation; Software; Visualization; Writing—review and editing. **Aleksandr Fedorov**: Investigation; Visualization; Methodology; Writing—original draft; Writing—review and editing. **David Sims**: Supervision; Writing—review and editing. **Jan Rehwinkel**: Conceptualization; Formal analysis; Supervision; Funding acquisition; Validation; Visualization; Writing—original draft; Writing—review and editing.

Source data underlying figure panels in this paper may have individual authorship assigned. Where available, figure panel/source data authorship is listed in the following database record: biostudies:S-SCDT-10_1038-S44319-026-00750-3.

## Disclosure and competing interests statement

The authors declare no competing interests.

