## [Peer Review File · EMBO Reports]

Single-cell analysis of signalling and transcriptional responses to type I interferons

Rachel Rigby, Kevin Rue-Albrecht, Aleksandr Fedorov, David Sims, and Jan Rehwinkel

Corresponding author: Jan Rehwinkel (jan.rehwinkel@imm.ox.ac.uk)

Review Timeline:

Transfer from Review Commons:	29th Aug 25
Editorial Decision:	6th Nov 25
Revision Received:	10th Dec 25
Editorial Decision:	22nd Jan 26
Revision Received:	6th Feb 26
Accepted:	6th Mar 26

Transaction Report:

This manuscript was transferred to EMBO Reports following peer review at Review Commons.

Note: With the exception of the correction of typographical or spelling errors that could be a source of ambiguity, letters and reports are not edited. Depending on transfer agreements, referee reports obtained elsewhere may or may not be included in this compilation. Referee reports are anonymous unless the Referee chooses to sign their reports.)

Review #1

1. Evidence, reproducibility and clarity:

Evidence, reproducibility and clarity (Required)

The study can be directly connected to a landmark paper in the field (Mostafavi et al. , Cell 2016). By comparison with this study, the authors use improved technologies to address the question if and how responses to type I IFN differ between human peripheral blood-derived cells types. In line with Mostafavi et al. the authors conclude that only a comparably low number of interferon-stimulated genes (ISG) is induced in all cell types and that considerable differences exist between cell types in the IFN-induced transcriptome. The authors address a second relevant aspect, whether and how the many different subtypes of type I IFN differ in the way they engage IFN signals to produce transcriptome changes. The data lead the authors to conclude that any differences are of quantitative rather than qualitative nature.

The authors' conclusions are based on a mass cytometry approach to phenotype STAT activation in different cell types, bulk RNA sequencing to study ISG expression in PBMC, and single cell sequencing to study ISG responses in individual cell types. The data are solid, clear and reproducible in biological replicates (eg different blood donors).

2. Significance:

Significance (Required)

While some of the data can be considered confirmatory, the comprehensive analysis of cell-type specificity and IFN-I subtype specificity advances the field and provides a reference for future analyses. The study is complete and there is no obvious lack of a critical experiment. The number of scientists interested in the multitude of open questions around type I IFN is large, thus the study is likely to attract a broad readership.

The biggest limitation is to my opinion the low sequencing depth of scRNAseq which is clearly the downside of this technology. Using 11 hematopoietic cell types and bulk RNA sequencing the total number of ISG was determined to be 975 by Mostafavi et al. and the core ISG numbered 166. This is in stark contrast to this studies' 10 core ISG. The authors limitations paragraph should discuss the fact that scRNAseq reduces the overall ISG number that can be analyzed.

A minor point concerns the 25 supplementary figures of the study. There must be a better way to support the conclusions with the necessary data.

3. How much time do you estimate the authors will need to complete the suggested revisions:

Estimated time to Complete Revisions (Required)

(Decision Recommendation)

Less than 1 month

Yes

Review #2

1. Evidence, reproducibility and clarity:

Evidence, reproducibility and clarity (Required)

The manuscript entitled "Single-cell analysis of signalling and transcriptional responses to type I interferon" by Rigby et al. examines the response to type I IFN subtypes in PBMCs using an integrative proteomics and transcriptomics approach. Some of the analysis could be deepened to provide better insights into what governs the magnitude of change in gene expression as well as the cell type-specific response to expression and generate more excitement for the study.

****Major Comments:****

1. Although the authors appropriately conclude that type I IFNs induce qualitatively similar, the response is not quantitatively similar. What elements in the promoters of ISGs make them more responsive to IFN subtypes? (PMID: 32847859) How do they relate to the activation of kinases by IFN subtypes? Are there distinct features that dictate differential responses in monocytes and lymphocytes?
2. Figure 2a, d-h - Consider using the same scale for all heatmaps. This will allow for comparison of pSTATs median expression. Consider increasing the range in the color scale as some of the subtle changes in STAT phosphorylation across subtypes are not well appreciated. This also applies to Supplementary figures related to Figure 2.

****Minor Comments:****

1. The title of subsections are a bit generic (e.g "Analysis of the signalling response to type I IFNs using mass cytometry". Consider updating them to reflect some of the findings from each analysis.
2. Figure 3 and S3 - Increase the heatmap scale to better appreciate changes in gene expression.
3. Consider combining panel a and b in figure S7 for better contrasts of the response to IFN α 1 or IFN β .
4. Figure 4 - The authors could visualize ISGs that are unique across IFN types or cell types.
5. The gene ontology analysis should be performed with higher statistical stringency to capture the most significant IFN responsive processes.

2. Significance:

Significance (Required)

Significance:

The authors provide an extensive compendium of cell type specific changes in response to type I IFN stimulation. They have created a public repository which extends the value of this dataset.

Audience:

This is a valuable resource for immunologists, virologists, and bioinformaticians.

3. How much time do you estimate the authors will need to complete the suggested revisions:

Estimated time to Complete Revisions (Required)

(Decision Recommendation)

Less than 1 month

Yes

Review #3

1. Evidence, reproducibility and clarity:

Evidence, reproducibility and clarity (Required)

Summary

Rigby and collaborators analyzed the signaling responses and changes in gene expression of human PBMCs stimulated with different IFN type I subtypes, using mass cytometry, bulk and single-cell RNA sequencing. Their study represents the first single-cell atlas of human PBMCs stimulated with five type I IFN subtypes. The generated datasets are useful resources for anyone interested in innate immunity. The data and the methods are well presented. We thus recommend publication.

Major comments:

Two of the key conclusions are not very convincing.

First, the authors claim that the magnitude of the responses varied between the 5 types of IFNs, however, as they point out in the 'limitation' paragraph, doses of the different IFNs were normalized using bioactivity. Knowing that this bioactivity is based on assays performed on A549 lung cells, this normalization likely induces a bias. How do the authors explain similar antiviral bioactivity but differing magnitudes of modulation of ISG expression? Would the authors expect the same differences of expression between the several IFNs tested in A549 cells? We thus recommend being very cautious when comparing magnitude of the response between the 5 types of IFNs.

Second, the qualitatively different responses to type I IFN subtypes claimed by the authors were not apparent. This seems true at the level of the bulk population (Fig. S10) but not at cell-type level (Fig. S15/S16). The authors state (line 311-312) that 'Consistent with our bulk RNAseq data, differences were again quantitative rather than qualitative' at the cell-type level. The response between cell types seems very different to us since a core set of only 10 ISGs are shared by all cell types and all 5 type I IFNs. Knowing that the expression of hundreds, sometimes thousands of genes, are induced by IFN, this seems like a rather small overlap (and thus qualitatively different responses). Fig S15 and S16 nicely illustrate that the responses are qualitatively different between cell-type. Please modify this conclusion accordingly.

No additional experiments are needed to support the claims. However, we believe that two

additional analyses could provide useful information.

The levels of IFNAR1 and IFNAR2 expressed at the plasma membrane probably vary between cell types and may thus influence the magnitude of the IFN response. While it would be difficult to measure these levels by flow cytometric analysis on the different cell types, could the authors extract information from their scRNAseq analysis on the expression level of IFNAR1/2 in all cell types? This would give a hint about potential differences in expression (and thus in magnitude).

Could the authors investigate further the expression of lncRNAs at the single-cell levels? It would be useful to also define a core set of lncRNAs that are shared between cell types and IFN subtypes. If such a core set does not exist (since lncRNAs are less conserved than coding genes), it would be nice to mention it.

****Minor comments:****

There is a typo in line 355 Fig.4C =>6C.

****Referees cross-commenting****

We agree with Reviewer 1 that the low sequencing depth of scRNAseq restricts the analysis and must be discussed in the 'limitation' paragraph. This would explain why the authors identified only 10 ISGs that are common to all cell types and all 5 IFN subtypes. Of note, as a comparison, Shaw et al (10.1371/journal.pbio.2004086) identified a core set of 90 ISGs that are upregulated upon IFN treatment in cells isolated mainly from kidney and skin of nine mammalian species ("core mammalian ISGs"). It is thus expected that stimulated blood cells isolated from a single mammalian species share more than 10 ISGs.

2. Significance:

Significance (Required)

Multiple single-cell RNAseq analysis of PBMCs, stimulated or not, have been previously performed in multiple contexts (for instance with PBMCs isolated from the blood of patients infected with influenza virus or SARS-CoV-2). The technical advance is thus limited.

However, the work represents a conceptual advance for the field since it provides the first single-cell atlas of PBMCs stimulated with five type-I IFN subtypes. The generated datasets represent a great resource for anyone interested in innate immunity (virologists, immunologists and cancerologists).

Of note, we are studying innate immunity in the context of RNA virus infection but we have no expertise on scRNA sequencing. We may thus have missed a flaw in the analyses.

3. How much time do you estimate the authors will need to complete the suggested revisions:

Estimated time to Complete Revisions (Required)

(Decision Recommendation)

Less than 1 month

Yes

Full Revision

Manuscript number: RC-2023-02191

Corresponding author: Jan Rehwinkel

1. General Statements

The authors wish to thank all three reviewers and the Review Commons team for carefully evaluating our study. We have addressed all points raised as detailed below.

We have thoroughly revised our bulk RNAseq analysis, which is now performed at the transcript level using the latest GENCODE release. We have updated Figure 3 and associated supplementary figures and tables. This change from gene to transcript level was important for accurate motif analysis as requested by reviewer 2: matching promoters to individual IFN-regulated transcripts – rather than aggregating all promoters per gene – avoids significant signal dilution. This strategy yields higher-resolution expression data and is biologically preferable. Indeed, several well characterised IFN-regulated RNAs (e.g., the ADAR1-202 transcript encoding the p150 isoform) originate from promoters located far from the constitutive promoters of their host genes. In our revised manuscript, we now provide in the new supplementary figure 13 the requested promoter motif analysis. Using two computational approaches – *de novo* motif search and analysis of a curated motif database – we find strong enrichment of interferon-stimulated response elements (ISREs) in promoters of type I IFN regulated transcripts. No other motifs reached similarly high levels of enrichment, and our analysis did not reveal differences between different type I IFNs. These new data show that all type I IFNs engage a common regulatory pathway, supporting our overall conclusion that different type I IFNs do not induce qualitatively different responses in PBMCs.

Regrettably, in the process of analysing the bulk RNAseq data at transcript level, we noticed that our original lncRNA analysis contained numerous false positives. Closer inspection showed that many “differentially expressed” LNCipedia models were likely not full-length transcripts and commonly shared a single IFN-induced set of exons that artificially inflated expression estimates for every overlapping model. To correct this issue, we replaced LNCipedia with the latest high-quality non-coding RNA catalogue from GENCODE, most entries of which were defined by full-length RNA sequencing [1]. We also tightened our filtering criteria and now report only transcripts that are robustly expressed in our dataset and are either classified as high-confidence by GENCODE or robustly supported at every splice junction by our RNAseq.

We hope our manuscript is sufficiently improved and suitable for publication. New or revised text is highlighted in green in our revised manuscript.

Reviewer #1

Evidence, reproducibility and clarity:

The study can be directly connected to a landmark paper in the field (Mostafavi et al. , Cell 2016). By comparison with this study, the authors use improved technologies to address the question if and how responses to type I IFN differ between human peripheral blood-derived cells types. In line with Mostafavi et al. the authors conclude that only a comparably low number of interferon-stimulated genes (ISG) is induced in all cell types and that considerable differences exist between cell types in the IFN-induced transcriptome. The authors address a second relevant aspect, whether and how the many different subtypes of type I IFN differ in the way they engage IFN signals to produce transcriptome changes. The data lead the authors to conclude that any differences are of quantitative rather than qualitative nature.

The authors' conclusions are based on a mass cytometry approach to phenotype STAT activation in different cell types, bulk RNA sequencing to study ISG expression in PBMC, and single cell sequencing to study ISG responses in individual cell types. The data are solid, clear and reproducible in biological replicates (eg different blood donors).

Significance:

While some of the data can be considered confirmatory, the comprehensive analysis of cell-type specificity and IFN-I subtype specificity advances the field and provides a reference for future analyses. The study is complete and there is no obvious lack of a critical experiment. The number of scientists interested in the multitude of open questions around type I IFN is large, thus the study is likely to attract a broad readership.

We thank the reviewer for her/his positive assessment of our study.

The biggest limitation is to my opinion the low sequencing depth of scRNAseq which is clearly the downside of this technology. Using 11 hematopoietic cell types and bulk RNA sequencing the total number of ISG was determined to be 975 by Mostafavi et al. and the core ISG numbered 166. This is in stark contrast to this studies' 10 core ISG. The authors limitations paragraph should discuss the fact that scRNAseq reduces the overall ISG number that can be analyzed.

Thank you for this valid comment. We amended the limitations paragraph as requested. We agree that the Mostafavi *et al.* 2016 Cell paper [2] is important but note that there are many differences to our study: Mostafavi *et al.* use mice, a seemingly very high IFN dose (10,000 Units) and microarrays (not RNAseq).

A minor point concerns the 25 supplementary figures of the study. There must be a better way to support the conclusions with the necessary data.

We agree that our supplementary materials are extensive. However, this is not unusual for studies reporting multiple large datasets. We would be delighted to organise our supplementary information differently in due course according to journal guidelines.

Reviewer #2

Evidence, reproducibility and clarity:

The manuscript entitled "Single-cell analysis of signalling and transcriptional responses to type I interferon" by Rigby et al. examines the response to type I IFN subtypes in PBMCs using an integrative proteomics and transcriptomics approach. Some of the analysis could be deepened to provide better insights into what governs the magnitude of change in gene expression as well as the cell type-specific response to expression and generate more excitement for the study.

We thank the reviewer for evaluating our study and the suggestions made.

Major Comments:

1. Although the authors appropriately conclude that type I IFNs induce qualitatively similar, the response is not quantitatively similar. What elements in the promoters of ISGs make them more responsive to IFN subtypes? (PMID: 32847859)

We thank the reviewer for the suggestion to study the promoters of genes regulated by type I IFNs. The analyses outlined below were performed by A. Fedorov, who is now a new co-author of our study. To investigate promoter features that might underlie the observed transcriptional responses across type I IFNs, we first performed a *de novo* motif search using STREME [3] on our bulk RNAseq dataset (Figure 3). Specifically, we compared the promoters of transcripts that were up- or down-regulated by each IFN subtype (e.g., IFN- β -induced) either with one another or with promoters of robustly expressed RNAs that remained unresponsive to any treatment. No significant motifs emerged from these comparisons, except when we compared promoters of IFN-induced transcripts to the background set of unresponsive RNAs. This comparison consistently yielded strong enrichment of interferon-stimulated response element (ISRE)-like motifs in the promoters of up-regulated RNAs (new Figure S13a).

Next, we conducted a complementary analysis using known transcription factor (TF) motifs from the JASPAR database [4]. We screened all promoters of annotated RNAs using clustered JASPAR motifs and Z-standardised motif scores relative to all high-confidence GENCODE RNAs, including those not expressed in PBMCs. We reasoned that TFs actively mediating IFN responses would likely bind promoters with high motif scores ($Z \geq 2$), while promoters with low scores ($Z \leq -1$) would represent an unregulated background. This approach produced two sets of RNAs per TF cluster: putatively regulated and unregulated. We then restricted each set to RNAs expressed in our dataset and associated each transcript with its estimated fold change in response to each type I IFN, regardless of statistical significance. Next, we compared median fold changes between the likely regulated and unregulated sets across all TF clusters and IFN subtypes (Figure S13b). Among all tested TF motifs, only the ISRE-like cluster showed strong and consistent associations with transcriptional changes across all IFN subtypes. We also observed statistically significant but much weaker associations for other TFs, including a known negative regulator of innate antiviral signaling, NRF1 [5]. However, effect sizes for these motifs were dwarfed by those of ISRE-like motifs, suggesting that no JASPAR TFs other than those within the ISRE-like cluster play a major role in PBMCs under our conditions. Overall, these findings support the idea that all type I IFNs engage a common regulatory pathway, differing primarily in the magnitude rather than the nature of their transcriptional effects.

How do they relate to the activation of kinases by IFN subtypes?

We did not analyse the activation of the canonical kinases (i.e., TYK2 and JAK1) downstream of IFNAR. This would be interesting and may be possible using phospho-specific antibodies to these kinases in our CyTOF setup. However, this would require a very large investment of time and resources to identify specific antibodies, optimise a new CyTOF staining panel and to acquire and analyse new datasets. We therefore believe this should be pursued as a separate future study.

Are there distinct features that dictate differential responses in monocytes and lymphocytes?

Following the computational approach described above, we applied STREME to identify DNA motifs that could distinguish promoters associated with monocyte- and lymphocyte-specific ISGs. Regrettably, this analysis did not yield any significant motifs, likely due in part to the limited number of genes in each category.

2. Figure 2a, d-h - Consider using the same scale for all heatmaps. This will allow for comparison of pSTATs median expression. Consider increasing the range in the color scale as some of the subtle changes in STAT phosphorylation across subtypes are not well appreciated. This also applies to Supplementary figures related to Figure 2.

Thank you for this suggestion. We tried using the same scale for all heatmaps. However, given that the values for pSTAT1 are higher than those for other pSTATs, the resulting heatmaps did not show differences for the other pSTATs well. We therefore decided to leave these panels unchanged. Please also note that Figures 2b and S3b provide comparison between pSTATs (and other markers) using the same scale.

Minor Comments:

1. The title of subsections are a bit generic (e.g "Analysis of the signalling response to type I IFNs using mass cytometry"). Consider updating them to reflect some of the findings from each analysis.

Thank you for this suggestion. We have amended sub-headers accordingly.

2. Figure 3 and S3 - Increase the heatmap scale to better appreciate changes in gene expression.

The scales have been enlarged for better visibility as requested.

3. Consider combining panel a and b in figure S7 for better contrasts of the response to IFN α 1 or IFN β .

Thank you for the suggestion. We combined these panels.

4. Figure 4 - The authors could visualize ISGs that are unique across IFN types or cell types.

Figure 5 and several accompanying supplementary figures already depict ISGs unique to IFN subtypes or cell types. Whilst we appreciate the suggestion, we prefer not to add additional figures to avoid redundancies.

5. The gene ontology analysis should be performed with higher statistical stringency to capture the most significant IFN responsive processes.

Thank you for this comment. We changed the presentation of the GO analysis in Fig S11 by sorting on p-value (instead of % of hits in category). We hope this shows more clearly that GO category enrichment amongst genes encoding IFN-induced transcripts had high statistical significance (log₁₀ p-values of about -5 or lower for many categories).

Significance:

The authors provide an extensive compendium of cell type specific changes in response to type I IFN stimulation. They have created a public repository which extends the value of this dataset.

Audience:

This is a valuable resource for immunologists, virologists, and bioinformaticians.

Thank you for these encouraging comments.

Reviewer #3

Evidence, reproducibility and clarity:

Summary

Rigby and collaborators analyzed the signaling responses and changes in gene expression of human PBMCs stimulated with different IFN type I subtypes, using mass cytometry, bulk and single-cell RNA sequencing. Their study represents the first single-cell atlas of human PBMCs stimulated with five type I IFN subtypes. The generated datasets are useful resources for anyone interested in innate immunity. The data and the methods are well presented. We thus recommend publication.

Thank you for your positive assessment of our work and for recommending publication.

Major comments:

Two of the key conclusions are not very convincing.

First, the authors claim that the magnitude of the responses varied between the 5 types of IFNs, however, as they point out in the 'limitation' paragraph, doses of the different IFNs were normalized using bioactivity. Knowing that this bioactivity is based on assays performed on A549 lung cells, this normalization likely induces a bias. How do the authors explain similar antiviral bioactivity but differing magnitudes of modulation of ISG expression? Would the authors expect the same differences of expression between the several IFNs tested in A549 cells? We thus recommend being very cautious when comparing magnitude of the response between the 5 types of IFNs.

We thank the reviewer for this important point and included the following reasoning in our discussion:

“An important technical consideration for our study was the normalisation of type I IFN doses used to treat cells (see also ‘Limitations of the study’ below). We relied on bioactivity (U/ml) that is measured by the manufacturer of recombinant type I IFNs using a cytopathic effect (CPE) inhibition assay. In brief, the lung cancer cell line A549 is treated with type I IFN and is infected with the cytopathic encephalomyocarditis virus (EMCV). Control cells not treated with IFN are killed by EMCV, whereas cells treated with sufficient IFN survive. How, then, is it possible that different type I IFNs induce differing magnitudes of STAT phosphorylation and ISG expression despite being used at the same bioactivity? Cell survival in the CPE inhibition assay may be due to one or a few ISGs. Indeed, single ISGs can mediate powerful antiviral defence. For example, MX1 is crucial for host defence against influenza A virus [6]. Thus, similar bioactivity of different IFNs in A549 cells against EMCV-triggered cell death may not reflect the breadth of effects on many ISGs. Moreover, IFN-induced survival of A549 cells following EMCV infection is a binary readout. Induction of the relevant ISG(s) mediating protection beyond a threshold required for cell survival is unlikely to register in this assay. Thus, similar antiviral bioactivity (in the CPE inhibition assay) and differing magnitudes of modulation of ISG expression (at transcriptome level) are compatible.”

We believe inclusion of this paragraph demonstrates an appropriate level of caution in our data interpretation. Further, we would expect to make similar observations if we were to apply transcriptomic analysis to A549 cells treated with different type I IFNs. However, given our focus

in this study on primary, normal cells, we decided not to pursue work with the transformed and lab adapted A549 cell line.

Second, the qualitatively different responses to type I IFN subtypes claimed by the authors were not apparent. This seems true at the level of the bulk population (Fig. S10) but not at cell-type level (Fig. S15/S16).

We believe there may be a misunderstanding here. In relation to Figure S10, we do not claim “qualitatively different responses to type I IFN subtypes”. Instead, we conclude that “differences in expression between the different type I IFNs were quantitative” (page 8; lines 229-230, now: 238-239). Moreover, Figures S15/S16 (now: S16/S17) do not refer to analyses of responses to different type I IFN subtypes.

The authors state (line 311-312) that 'Consistent with our bulk RNAseq data, differences were again quantitative rather than qualitative' at the cell-type level. The response between cell types seems very different to us since a core set of only 10 ISGs are shared by all cell types and all 5 type I IFNs. Knowing that the expression of hundreds, sometimes thousands of genes, are induced by IFN, this seems like a rather small overlap (and thus qualitatively different responses). Fig S15 and S16 nicely illustrate that the responses are qualitatively different between cell-type. Please modify this conclusion accordingly.

Thank you for highlighting this. The statement in lines 311-312 does not refer to differences between cell types but to differences between type I IFN subtypes. We are sorry this was not clear and changed this sentence (now lines 357-358). Furthermore, we have made it clearer in the revised text that qualitative differences were observed between cell types (e.g. lines 329 and 350-352).

No additional experiments are needed to support the claims. However, we believe that two additional analyses could provide useful information.

The levels of IFNAR1 and IFNAR2 expressed at the plasma membrane probably vary between cell types and may thus influence the magnitude of the IFN response. While it would be difficult to measure these levels by flow cytometric analysis on the different cell types, could the authors extract information from their scRNAseq analysis on the expression level of IFNAR1/2 in all cell types? This would give a hint about potential differences in expression (and thus in magnitude).

We analysed *IFNAR1/2* transcript levels in our scRNAseq dataset (Figure R1 below). Unfortunately, for many cells, *IFNAR1* and *IFNAR2* transcripts were not detected (see width of violin plots at zero), probably due to low sequencing depth inherent to scRNAseq analysis. We therefore prefer not to draw conclusions from these data.

Could the authors investigate further the expression of lncRNAs at the single-cell levels? It would be useful to also define a core set of lncRNAs that are shared between cell types and IFN subtypes. If such a core set does not exist (since lncRNAs are less conserved than coding genes), it would be nice to mention it.

Thank you for this suggestion. The expression of lncRNAs is generally lower than protein-coding genes, resulting in high drop-out rates in 10X datasets. Indeed, Zhao *et al.* comment that “current development of single-cell technologies may not yet be optimized for lncRNA detection

and quantification" [7]. We only detected a small number of lncRNAs in our scRNAseq analysis, and only four lncRNAs were significantly differentially expressed between cell types. We thus could not perform a meaningful analysis of lncRNAs in our scRNAseq dataset. This is now mentioned in the limitations paragraph at the end of the manuscript.

Minor comments:

There is a typo in line 355 Fig.4C =>6C.

Thank you for spotting this.

Referees cross-commenting

We agree with Reviewer 1 that the low sequencing depth of scRNAseq restricts the analysis and must be discussed in the 'limitation' paragraph. This would explain why the authors identified only 10 ISGs that are common to all cell types and all 5 IFN subtypes. Of note, as a comparison, Shaw et al (10.1371/journal.pbio.2004086) identified a core set of 90 ISGs that are upregulated upon IFN treatment in cells isolated mainly from kidney and skin of nine mammalian species ("core mammalian ISGs"). It is thus expected that stimulated blood cells isolated from a single mammalian species share more than 10 ISGs.

We amended the limitations section as requested. Shaw *et al.* [8] used a single type I IFN (universal or IFN α , depending on species) at a very high dose (1000 U/ml). Taken together with the use of bulk RNAseq in this study, it is unsurprising that our work identified fewer core ISGs. We believe our small list of core ISGs is nonetheless both a high confidence and a high utility set of ISGs: these genes are induced by multiple type I IFNs, in all major cell types in blood and their regulation can be measured even when sequencing depth is low.

Significance (Required)

Multiple single-cell RNAseq analysis of PBMCs, stimulated or not, have been previously performed in multiple contexts (for instance with PBMCs isolated from the blood of patients infected with influenza virus or SARS-CoV-2). The technical advance is thus limited.

However, the work represents a conceptual advance for the field since it provides the first single-cell atlas of PBMCs stimulated with five type-I IFN subtypes. The generated datasets represent a great resource for anyone interested in innate immunity (virologists, immunologists and cancerologists).

Of note, we are studying innate immunity in the context of RNA virus infection but we have no expertise on scRNA sequencing. We may thus have missed a flaw in the analyses.

We thank the reviewer for their positive assessment of the advances of our study and the value of our IFN resource.

Figure R1. *IFNAR1/2* expression in scRNAseq data. Violin plots showing expression of *IFNAR1* (A,C) or *IFNAR2* (B,D) in different cell types. In (A,B), data were pooled across conditions. In (C,D), data are shown separately for unstimulated control cells and cells stimulated with different type I IFNs.

References

1. Kaur G, Perteghella T, Carbonell-Sala S, Gonzalez-Martinez J, Hunt T, Madry T, et al. GENCODE: massively expanding the lncRNA catalog through capture long-read RNA sequencing. *bioRxiv*. 2024. Epub 20241031. doi: 10.1101/2024.10.29.620654. PubMed PMID: 39554180; PubMed Central PMCID: PMCPMC11565817.
2. Mostafavi S, Yoshida H, Moodley D, LeBoite H, Rothamel K, Raj T, et al. Parsing the Interferon Transcriptional Network and Its Disease Associations. *Cell*. 2016;164(3):564-78. Epub 2016/01/30. doi: 10.1016/j.cell.2015.12.032. PubMed PMID: 26824662; PubMed Central PMCID: PMCPMC4743492.
3. Bailey TL. STREME: accurate and versatile sequence motif discovery. *Bioinformatics*. 2021;37(18):2834-40. doi: 10.1093/bioinformatics/btab203. PubMed PMID: 33760053; PubMed Central PMCID: PMCPMC8479671.
4. Rauluseviciute I, Riudavets-Puig R, Blanc-Mathieu R, Castro-Mondragon JA, Ferenc K, Kumar V, et al. JASPAR 2024: 20th anniversary of the open-access database of transcription factor binding profiles. *Nucleic acids research*. 2024;52(D1):D174-D82. doi: 10.1093/nar/gkad1059. PubMed PMID: 37962376; PubMed Central PMCID: PMCPMC10767809.
5. Zhao T, Zhang J, Lei H, Meng Y, Cheng H, Zhao Y, et al. NRF1-mediated mitochondrial biogenesis antagonizes innate antiviral immunity. *The EMBO journal*. 2023;42(16):e113258. Epub 20230706. doi: 10.15252/embj.2022113258. PubMed PMID: 37409632; PubMed Central PMCID: PMCPMC10425878.
6. Grimm D, Staeheli P, Hufbauer M, Koerner I, Martinez-Sobrido L, Solorzano A, et al. Replication fitness determines high virulence of influenza A virus in mice carrying functional Mx1 resistance gene. *Proceedings of the National Academy of Sciences of the United States of America*. 2007;104(16):6806-11. Epub 20070410. doi: 10.1073/pnas.0701849104. PubMed PMID: 17426143; PubMed Central PMCID: PMCPMC1871866.
7. Zhao X, Lan Y, Chen D. Exploring long non-coding RNA networks from single cell omics data. *Comput Struct Biotechnol J*. 2022;20:4381-9. Epub 20220804. doi: 10.1016/j.csbj.2022.08.003. PubMed PMID: 36051880; PubMed Central PMCID: PMCPMC9403499.
8. Shaw AE, Hughes J, Gu Q, Behdenna A, Singer JB, Dennis T, et al. Fundamental properties of the mammalian innate immune system revealed by multispecies comparison of type I interferon responses. *PLoS Biol*. 2017;15(12):e2004086. Epub 2017/12/19. doi: 10.1371/journal.pbio.2004086. PubMed PMID: 29253856.

Dear Dr. Rehwinkel

Thank you for the submission of your research manuscript to our journal. I apologize for the delay in handling it, but we have now received the reports from the two referees I have asked to assess it. As you will see, both of them consider their concerns adequately addressed and recommend publication.

I will now need you to revise your study to match the formatting requirements we have. I list all relevant points in the section GENERAL FORMATTING GUIDELINES. Once the revised manuscript has been submitted, we will perform a number of quality control and image integrity checks. Please note that we will also need Source Data (point 8).

To speed up the process, I note here already some specific points that need to be addressed:

- The Data Availability section needs links that resolve directly to the deposited datasets.
- The reuse of data from reference (69) should be acknowledged with a Data reference (see point 11).
- The statement "Any additional information required to reanalyse the data reported in this paper is available from the lead contact upon request." Should be removed from the Data Availability section.
- Author contributions need to be removed from the manuscript text. It is sufficient to specify them in the online manuscript tracking system. This information will be retrieved and typeset into the article.
- All figures need to be uploaded as individual files. Only the legends remain in the article file.
- References need to be alphabetical, not numerical.
- Supplemental files can either be combined to an Appendix PDF or be supplied as Expanded View items.
- Supplemental tables that report complex datasets should be converted to Datasets (Dataset EV#). These need a legend in a separate tab of the .xls file.

GENERAL FORMATTING GUIDELINES:

2) individual production quality figure files as .eps, .tif, .jpg (one file per figure).

Please download our Figure Preparation Guidelines (figure preparation pdf) from our Author Guidelines pages <https://www.embopress.org/page/journal/14693178/authorguide> for more info on how to prepare your figures.

4) a complete author checklist, which you can download from our author guidelines (). Please insert information in the checklist that is also reflected in the manuscript. The completed author checklist will also be part of the RPF.

5) Please note that all corresponding authors are required to supply an ORCID ID for their name upon submission of a revised manuscript (). Please find instructions on how to link your ORCID ID to your account in our manuscript tracking system in our Author guidelines

()

6) We replaced Supplementary Information with Expanded View (EV) Figures and Tables that are collapsible/expandable online. A maximum of 5 EV Figures can be typeset. EV Figures should be cited as 'Figure EV1, Figure EV2' etc... in the text and their respective legends should be included in the main text after the legends of regular figures.

7) Before submitting your revision, primary datasets (and computer code, where appropriate) produced in this study need to be

deposited in an appropriate public database (see <
<https://www.embopress.org/page/journal/14693178/authorguide#dataavailability>>).

The accession numbers and database should be listed in a formal "Data Availability " section (placed after Materials & Method) that follows the model below (see also <
<https://www.embopress.org/page/journal/14693178/authorguide#dataavailability>>). Please note that the Data Availability Section is restricted to new primary data that are part of this study.

Data availability

Additional information on source data and instruction on how to label the files are available

10) Figure legends and data quantification:

- the name of the statistical test used to generate error bars and P values,
 - the EXACT p-values,
 - the number (n) of independent experiments (please specify technical or biological replicates) underlying each data point,
 - the nature of the bars and error bars (s.d., s.e.m.)
- If the data are obtained from n {less than or equal to} 5, show the individual data points in addition to the SD or SEM.
 - If the data are obtained from n {less than or equal to} 2, use scatter blots showing the individual data points.

11) Our journal encourages inclusion of *data citations in the reference list* to directly cite datasets that were re-used and obtained from public databases. Data citations in the article text are distinct from normal bibliographical citations and should directly link to the database records from which the data can be accessed. In the main text, data citations are formatted as follows: "Data ref: Smith et al, 2001" or "Data ref: NCBI Sequence Read Archive PRJNA342805, 2017". In the Reference list, data citations must be labeled with "[DATASET]". A data reference must provide the database name, accession number/identifiers and a resolvable link to the landing page from which the data can be accessed at the end of the reference. Further instructions are available at .

12) All Materials and Methods need to be described in the main text using our 'Structured Methods' format. According to this format, the Methods section includes a Reagents and Tools Table (listing key reagents, experimental models, software and relevant equipment and including their sources and relevant identifiers) followed by a Methods and Protocols section describing the methods, ideally using a step-by-step protocol format. The aim is to facilitate adoption of the methodologies across labs. Please download and fill our Reagents and Tools Table template (.docx), which you can find in our author guidelines:

13) As part of the EMBO publication's Transparent Editorial Process, EMBO Reports publishes online a Review Process File to accompany accepted manuscripts. This File will be published in conjunction with your paper and will include the referee reports, your point-by-point response and all pertinent correspondence relating to the manuscript.

Kind regards,

=====

Referee #1:

The authors have made an adequate adjustment to my major comment. The manuscript has been further improved by the motif search in promoters of IFN-regulated transcripts. The finding that there is no significantly enriched motif other than the ISRE is interesting. The authors do not mention the possibility that the quantitative differences in DET might thus result from differences in the affinities of STAT or and/or IRF complexes for the respective sites.

Referee #2:

The authors have addressed the reviewer's concerns/suggestions adequately.

Referee #1:

The authors have made an adequate adjustment to my major comment. The manuscript has been further improved by the motif search in promoters of IFN-regulated transcripts. The finding that there is no significantly enriched motif other than the ISRE is interesting. The authors do not mention the possibility that the quantitative differences in DET might thus result from differences in the affinities of STAT or and/or IRF complexes for the respective sites.

Referee #2:

The authors have addressed the reviewer's concerns/suggestions adequately.

Rev_Com_number: RC-2023-02191

New_manu_number: EMBOR-2025-62649V1-T

Corr_author: Rehwinkel

Title: Single-cell analysis of signalling and transcriptional responses to type I interferons

Response to editorial and reviewers' comments

Manuscript Number: **EMBOR-2025-62649V1-T**

" Single-cell analysis of signalling and transcriptional responses to type I interferons "

Rachel E. Rigby, Kevin Rue-Albrecht, Aleksandr Fedorov, David Sims and Jan Rehwinkel

The authors wish to thank both reviewers and the editor for carefully evaluating our study. We were delighted to hear that we are invited to submit a revised version of our manuscript. We have now revised our manuscript, considering specific editorial points, general formatting guidelines, and reviewer's comments. We hope our manuscript is sufficiently improved for publication in *EMBO Reports*. New or revised text is highlighted in blue in our revised manuscript.

Point-by-point reply

Specific editorial points:

- The Data Availability section needs links that resolve directly to the deposited datasets.

We included links as requested.

- The reuse of data from reference (69) should be acknowledged with a Data reference (see point 11).

This has been changed as requested.

- The statement "Any additional information required to reanalyse the data reported in this paper is available from the lead contact upon request." Should be removed from the Data Availability section.

This sentence is now deleted.

- Author contributions need to be removed from the manuscript text. It is sufficient to specify them in the online manuscript tracking system. This information will be retrieved and typeset into the article.

This is now deleted from the manuscript text file.

- All figures need to be uploaded as individual files. Only the legends remain in the article file.

This has been done.

- References need to be alphabetical, not numerical.

We updated the references accordingly.

- Supplemental files can either be combined to an Appendix PDF or be supplied as Expanded View items.

We combined supplemental files into an Appendix PDF.

- Supplemental tables that report complex datasets should be converted to Datasets (Dataset EV#). These need a legend in a separate tab of the .xls file..

We renamed our supplemental tables to Datasets EV and include legends as instructed.

General formatting:

All source data for the main figures have been deposited in public data repositories and are freely available (see data availability section in the main manuscript). We thus do not provide source data files.

Reviewer #1:

The authors have made an adequate adjustment to my major comment. The manuscript has been further improved by the motif search in promoters of IFN-regulated transcripts. The finding that there is no significantly enriched motif other than the ISRE is interesting. The authors do not mention the possibility that the quantitative differences in DET might thus result from differences in the affinities of STAT or and/or IRF complexes for the respective sites.

We thank the reviewer for evaluating our study again. As suggested by the reviewer, we now mention that: “It is possible that quantitative differences result from differences in affinities or half-lives of the STAT and/or IRF complexes formed after engagement of IFNAR by different type I IFNs.”

Reviewer #2:

The authors have addressed the reviewer's concerns/suggestions adequately.

We thank the reviewer for evaluating our study again.

Additional changes:

Following a discussion at a recent conference with two senior scientists in the field (Art Krieg and Michael Lotze), we include three additional references¹⁻³ and two additional sentences in the discussion:

Moreover, clinical trials demonstrated distinct therapeutic effects. For example, IFN- α 2 is effective in treating some viral diseases and some forms of cancer, whereas IFN- β is used to treat multiple sclerosis (Ioannou & Isenberg, 2000).

We also note that quantitative differences, for example the particularly strong effects of IFN- β , may result in functionally distinct outcomes, including different clinical responses to treatment with different recombinant type I IFNs (Ioannou & Isenberg, 2000).

These additions do not change the conclusions of our study; they simply provide additional context by referring the reader to clinical use of type I IFNs as therapeutics.

References

1. Ioannou, Y., and Isenberg, D.A. (2000). Current evidence for the induction of autoimmune rheumatic manifestations by cytokine therapy. *Arthritis Rheum* *43*, 1431-1442. [10.1002/1529-0131\(200007\)43:7<1431::AID-ANR3>3.0.CO;2-E](https://doi.org/10.1002/1529-0131(200007)43:7<1431::AID-ANR3>3.0.CO;2-E).
2. Hilkens, C.M., Schlaak, J.F., and Kerr, I.M. (2003). Differential responses to IFN-alpha subtypes in human T cells and dendritic cells. *J Immunol* *171*, 5255-5263. [10.4049/jimmunol.171.10.5255](https://doi.org/10.4049/jimmunol.171.10.5255).
3. Foster, G.R., Masri, S.H., David, R., Jones, M., Datta, A., Lombardi, G., Runkell, L., de Dios, C., Sizing, I., James, M.J., and Marelli-Berg, F.M. (2004). IFN-alpha subtypes differentially affect human T cell motility. *J Immunol* *173*, 1663-1670. [10.4049/jimmunol.173.3.1663](https://doi.org/10.4049/jimmunol.173.3.1663).

Manuscript number: EMBOR-2025-62649V2

Title: Single-cell analysis of signalling and transcriptional responses to type I interferons

Author(s): Rachel Rigby, Kevin Rue-Albrecht, Aleksandr Fedorov, David Sims, and Jan Rehwinkel

Dear Dr. Rehwinkel

Thank you for your patience while we have reviewed your revised manuscript from the editorial side. As I informed you in my last letter, we perform a number of quality and data checks on revised manuscripts, and based on these, I need you to address a few minor issues and corrections, before I can officially accept the manuscript for publication:

- Order of manuscript sections:
- The Figure legends need to go after the References.
- The Disclosure and competing interest statement needs to be placed after Acknowledgments.

- You have provided only one funder in the online submission system, yet the Acknowledgment section lists a few more: Oxford Single Cell Biology Consortium (OSCBC), Clarendon Fund, the Radcliffe Department of Medicine and Keble College. This work was funded by the UK Medical Research Council [MRC core funding of the MRC Human Immunology Unit. It is important that the funding information in the manuscript and the system is congruent. Note that the information from the system is forwarded to the production team and ultimately to PubMed.

- The panels in the figures need to be labeled with capital letters (A, B etc).

- Figure callouts S9 should be Appendix Figure S9. The same applies for Appendix Figures S8c and S12. Make sure that all Supplementary Figures are called out as "Appendix Figure S#".

- The table of content of the Appendix needs all items listed separately with their page numbers.
- The panels should be labeled with capital letters (A instead of a).

- Appendix Figure S1C is too small to read the text on the individual violin blots. I suggest to split this figure into two so that you can show these subpanels in larger size.
- Appendix Figure S2C, S7, S8, S12, S13, S14, S15 and actually most of the Appendix figures have rather low resolution. In Appendix Figure S22 and S23 e.g., it is not possible to read the individual gene names. Please provide all figures in higher resolution.

- Please provide the Highlights summary text and bullet points as separate .docx file and the graphical abstract as a separate file (in jpeg, TIFF or png format: 550 pixels wide x 200-600 pixels high).

- The callouts to the Key Resources Table in line 579 and 612 should be corrected to Reagents and Tools table.

- Datasets EV14 and EV15 are not complex data and should either be provided as Table EV1 and EV2 or added to the Reagents and Tools table.

- Please address the following comments in the figure legends:
 - a) Please indicate the statistical test used for data analysis in the legends of figures 3A, B
 - b) Please provide information related to n in the legends of figures 3A, B; 5B

- Please write the abstract in present tense.

Once you have made these minor revisions, please use the following link to submit your corrected manuscript:

Link Not Available

If all remaining corrections have been attended to, you will then receive an official decision letter from the journal accepting your manuscript for publication in the next available issue of EMBO reports. This letter will also include details of the further steps you need to take for the prompt inclusion of your manuscript in our next available issue.

Thank you for your contribution to EMBO reports.

Kind regards,

Martina

The authors addressed the remaining editorial issues.

Jan Rehwinkel
University of Oxford
Radcliffe Department of Medicine
JR Hospital, WIMM, HIU
Headley Way
Oxford, Oxfordshire OX3 9DS
United Kingdom

Dear Dr. Rehwinkel,

I am very pleased to accept your manuscript for publication in the next available issue of EMBO reports. Thank you for your contribution to our journal.

You may qualify for financial assistance for your publication charges - either via a Springer Nature fully open access agreement or an EMBO initiative. Check your eligibility: <https://link.springer.com/journal/44319/how-to-publish-with-us>

Yours sincerely,

>>> Please note that it is EMBO Reports policy for the transcript of the editorial process (containing referee reports and your response letter) to be published as an online supplement to each paper. If you do NOT want this, you will need to inform the Editorial Office via email immediately. More information is available here: <https://link.springer.com/partners/embo-press/editorial-policies#Peer%20review>